# PDAgent: An LLM-Driven Autonomous Agent Framework Towards *In Silico* Protein Design via Directed Mutation

**Song Ouyang** [1]   **Zhijie Dong** [1]   **Yong Luo** [1]   **Kehua Su** [1]   **Huangxuan Zhao** [1]   **Miaojing Shi** [2]   **Bo Du** [1]

## Abstract

Computational protein design holds immense promise across diverse domains, but existing approaches face significant challenges: traditional physics-based methods require substantial domain expertise, while emerging deep learning methods often rely on restricted functional ontologies, struggle to bridge the semantic gap between text and protein sequences, or lack closed-loop optimization mechanisms. In this paper, we present PDAgent, an LLM-driven autonomous agent framework that enables *in silico* protein design through template-based directed mutation. Our framework accepts natural language specifications of desired protein properties and employs a ReAct-style reasoning loop comprising five phases: THINK, PLAN, ACT, OBSERVE, and REFLECT. PDAgent integrates template retrieval, conservation-aware mutation strategies, and domain-specific computational tools for property optimization across seven biophysical dimensions. Experiments on 100 diverse protein design tasks demonstrate that PDAgent achieves a 91.86% average constraint satisfaction rate with high structural quality (mean pLDDT 87.69), substantially outperforming both direct LLM generation and specialized deep learning methods. We provide the source code at `https://github.com/Gift-OYS/PDAgent`.

---

[1]School of Computer Science, National Engineering Research Center for Multimedia Software and Hubei Key Laboratory of Multimedia and Network Communication Engineering, Wuhan University, China [2]College of Electronic and Information Engineering, Tongji University, China. Correspondence to: Yong Luo <luoyong@whu.edu.cn>, Kehua Su <skh@whu.edu.cn>, Huangxuan Zhao <zhao_huangxuan@sina.com>.

*Proceedings of the $43^{rd}$ International Conference on Machine Learning*, Seoul, South Korea. PMLR 306, 2026. Copyright 2026 by the author(s).

## 1. Introduction

Proteins serve as fundamental building blocks of biological systems, with amino acid sequences dictating three-dimensional structures that determine specific biological activities (Dill & MacCallum, 2012). Protein design, which seeks to engineer sequences that realize desired functions or structures, is central to applications in biomedicine (Kintzing et al., 2016; Dauparas et al., 2022), industrial biotechnology (Miller et al., 2022; Buller et al., 2023), and environmental remediation (Kan et al., 2016; Lu et al., 2022). However, this inverse-engineering task is highly challenging: the combinatorial sequence space grows exponentially ($20^{100}$ for a 100-residue protein) (Tretyachenko et al., 2017), and sequence, structure, function, and physicochemical properties are coupled through complex relationships (Dill & MacCallum, 2012; Listov et al., 2024). Conventional computational approaches rely on physics-based energy functions (Leaver-Fay et al., 2011; Leman et al., 2020) to iteratively optimize sequences, but suffer from low efficiency, limited success rates, substantial computational demands, and the requirement for extensive expert configuration (Pan et al., 2022; Pan & Kortemme, 2021).

As deep learning advances protein design, researchers have explored diverse strategies for controllable generation, yet each faces distinct limitations. *Structure-based methods* (Dauparas et al., 2022; Hsu et al., 2022; Gao et al., 2023; Zheng et al., 2023) map 3D coordinates to amino acid distributions but provide limited functional control. *Function-conditioned methods* (Madani et al., 2023; Rao et al., 2021; Hayes et al., 2025) learn evolutionary patterns from functional annotations (e.g., GO terms (The Gene Ontology Consortium, 2026)), yet often produce hallucinated sequences and depend on predefined ontologies. *Text-conditioned methods* (Dai et al., 2024; Liu et al., 2025; Guo et al., 2025; Wang et al., 2025) align textual inputs with protein sequences but are constrained by cross-modal semantic gaps. *Large Language Model (LLM)-based agent systems* (Ghafarollahi & Buehler, 2024; Huang et al., 2025; Li et al., 2025) invoke external tools for complex tasks, yet typically lack mechanisms for iterative, closed-loop optimization.

To address these limitations, we propose PDAgent, an LLM-driven protein design framework inspired by ReAct (Yao

et al., 2022) that provides iteratively self-refining protein generation. PDAgent leverages LLM capabilities to interpret complex user requirements, extracting constraints (e.g., thermal stability, solubility, pH stability) and keywords for querying candidate templates from databases (Ahmad et al., 2025; UniProt, 2025). To ensure functional reliability, we incorporate conservation analysis via multiple sequence alignment (Ashkenazy et al., 2016; Ingles-Prieto et al., 2013) to identify and protect highly conserved residues during mutation. The framework implements a five-phase iterative cycle consisting of THINK, PLAN, ACT, OBSERVE, and RE-FLECT, forming a closed-loop workflow that decomposes protein design into reasoning and decision-making steps. After each mutation round, the model evaluates structural and functional changes in real time and updates its optimization strategy, enabling directed mutations in variable regions while preserving core functional domains.

Our contributions are summarized as follows:

- We propose PDAgent, an LLM-driven autonomous agent framework for natural language-guided protein design that enables iterative, closed-loop sequence optimization through a five-phase ReAct cycle.

- We integrate conservation analysis with stability-aware mutation filtering to ensure generated sequences maintain structural integrity while satisfying user-specified design constraints.

- We demonstrate through experiments on 100 diverse design tasks that PDAgent achieves 91.86% average constraint satisfaction rate with high mean pLDDT, substantially outperforming both direct LLM generation and specialized design methods.

## 2. Related Work

**Computational Protein Design.** Protein design can be viewed as an inverse problem: given a target function or intended structure, one seeks to engineer sequences that realize the desired characteristics. While traditional physics-based energy optimization methods, such as Rosetta (Leaver-Fay et al., 2011; Leman et al., 2020), have been widely used, their high time cost and low computational efficiency have driven a shift toward deep learning approaches. Structure-to-sequence methods, represented by ProteinMPNN (Dauparas et al., 2022) and ESM-IF (Hsu et al., 2022), predict amino acid sequences from backbone structures using graph neural networks and evolutionary information. Other approaches such as PiFold (Gao et al., 2023), UniIF (Gao et al., 2024b), KWDesign (Gao et al., 2024a), and diffusion-based methods (Yi et al., 2023; Ho et al., 2020; Austin et al., 2021) further advance protein inverse folding through diverse architectural innovations. In parallel, protein language models

(PLMs) provide powerful evolutionary priors for function-conditioned generation: ProGen (Madani et al., 2023), MSA Transformer (Rao et al., 2021), and ESM3 (Hayes et al., 2025) leverage masked or autoregressive objectives. Text-conditioned methods, including ProtDAT (Guo et al., 2025) and Pinal (Dai et al., 2024), align natural language with protein sequences via multimodal learning. More recently, LLM-based agent approaches such as DrugPilot (Li et al., 2025) invoke external tools for complex design tasks. Despite these advances, reliably steering generation toward user-defined functional objectives while maintaining structural plausibility remains an open challenge.

**LLMs for Scientific Discovery.** LLMs have emerged as a powerful foundation for scientific discovery due to their rich knowledge representations and strong reasoning capabilities (Wei et al., 2022; Kojima et al., 2022). By representing molecular SMILES strings, gene sequences, and protein sequences as tokenized inputs, LLMs demonstrate broad generalization across scientific domains. MolT5 (Edwards et al., 2022) enables effective cross-modal translation between molecular representations and natural language. BioGPT (Luo et al., 2022) leverages domain-specific pretraining for question answering and relation extraction, while Med-PaLM (Singhal et al., 2023) improves medical question answering through instruction tuning. SciAgents (Ghafarollahi & Buehler, 2025) demonstrates that multi-agent LLM systems can autonomously generate and refine scientific hypotheses by integrating ontological knowledge. ProteinDT (Liu et al., 2025) employs multimodal contrastive learning to align text and protein-sequence representations for text-guided generation, and ProtChatGPT (Wang et al., 2025) adopts progressive protein-language learning for interactive protein understanding and design. These advances demonstrate that LLMs can effectively bridge the semantic gap between human intent and scientific representations across diverse domains.

**LLM-based Autonomous Agents.** Reasoning paradigms including Chain-of-Thought (Wei et al., 2022; Kojima et al., 2022), Tree-of-Thought (Yao et al., 2023), ReAct (Yao et al., 2022), and Reflexion (Shinn et al., 2023), combined with tool-use capabilities (Schick et al., 2023; Qin et al., 2024), have substantially enhanced LLMs' ability to decompose complex tasks and interact with dynamic environments. Scientific applications span multiple domains: ChemCrow (Bran et al., 2024) for chemical synthesis planning, and Biomni (Huang et al., 2025) for biomedical research workflows. In protein design, ProtAgents (Ghafarollahi & Buehler, 2024) and ProtChat (Huang et al., 2024) have explored agent-assisted workflows with encouraging results. However, these existing approaches typically lack fully automated, closed-loop structured pipelines spanning design-validation-refinement, which limits their effectiveness for multi-objective optimization scenarios. This moti-

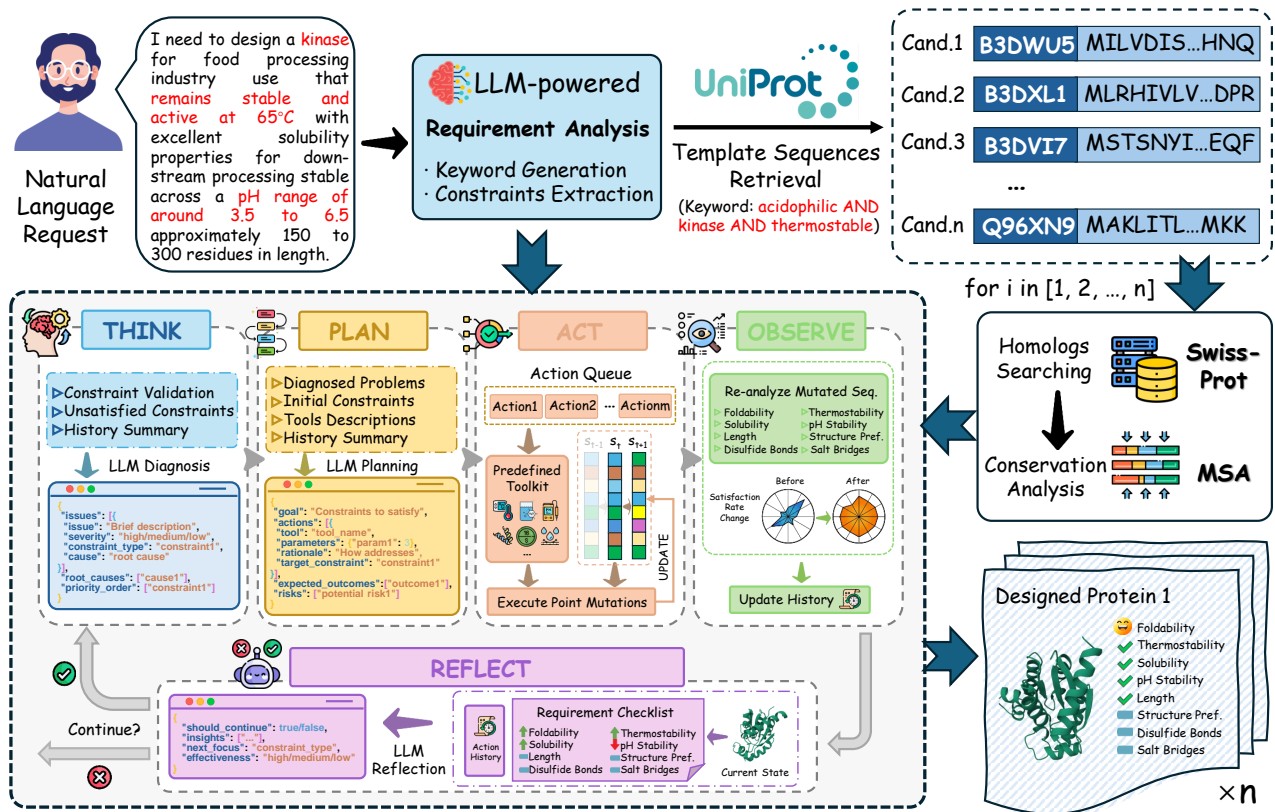

*Figure 1.* Overview of the PDAgent framework. The system processes natural language requests through six stages: (1) LLM-powered requirement analysis extracts constraints and keywords; (2) candidate generation retrieves template sequences from UniProt; (3) conservation analysis identifies protected positions via BLAST and MSA; (4) the ReAct optimization loop iteratively refines sequences through THINK, PLAN, ACT, OBSERVE, and REFLECT phases with shared history memory; (5-6) final analysis and output generation produce protein files.

vates a tighter integration of LLM reasoning with iterative property evaluation and sequence mutation, which is the direction we pursue in this work.

## 3. Method

### 3.1. Problem Formulation

Given a natural language design request $R$ describing desired protein properties, our design problem seeks an optimized sequence $\mathbf{s}^*$ that maximizes constraint satisfaction while maintaining structural viability. Let $\mathcal{A} = \{A_1, A_2, \ldots, A_{20}\}$ denote the standard amino acid alphabet, and let a protein sequence of length $L$ be represented as $\mathbf{s} = (s_1, s_2, \ldots, s_L) \in \mathcal{A}^L$. We extract a set of constraints $\mathcal{C} = \{c_1, c_2, \ldots, c_K\}$ from $R$, and define the optimization objective as maximizing the constraint satisfaction rate (CSR):

$$\mathbf{s}^* = \arg\max_{\mathbf{s} \in \mathcal{A}^L} \mathrm{CSR}(\mathbf{s}, \mathcal{C}) = \arg\max_{\mathbf{s}} \frac{1}{K} \sum_{i=1}^{K} \sigma_i(\mathbf{s}, c_i) \quad (1)$$

where $\sigma_i : \mathcal{A}^L \times \mathcal{C} \to \{0, 1\}$ is a constraint-specific binary satisfaction function that evaluates whether sequence $\mathbf{s}$ satisfies constraint $c_i$.

We define a unified representation for protein design constraints. Each constraint $c \in \mathcal{C}$ is specified as a tuple $c = (\tau, v, \gamma, r_{\min}, r_{\max})$, where $\tau \in \mathcal{T}$ is the constraint type from a predefined taxonomy, $v$ is the target value, $\gamma \in \{\texttt{geq}, \texttt{eq}, \texttt{range}, \texttt{bool}\}$ is the comparison operator, and $[r_{\min}, r_{\max}]$ defines acceptable bounds for range-type constraints. The satisfaction function $\sigma_i$ is then defined according to the comparison operator $\gamma$ (detailed definitions in Appendix A).

Our constraint taxonomy $\mathcal{T}$ covers seven biophysical dimensions essential for functional protein design: $\texttt{thermostability}$ (thermal stability at target temperature, validated by estimated melting temperature $T_m$), $\texttt{solubility}$ (assessed via GRAVY score), $\texttt{pH stability}$ (evaluated through isoelectric point matching), $\texttt{length}$ (number of amino acids), $\texttt{structure preference}$ ($\alpha$-helix and $\beta$-sheet fractions), $\texttt{disulfide bonds}$ (cysteine pair requirements),

**Algorithm 1** PDAgent Workflow

---

**Require:** Natural language request $R$, max iterations $T_{\max}$, thresholds $\theta_{\text{cons}}, \theta_{\text{ddG}}$

**Ensure:** Optimized protein sequence $\mathbf{s}^*$

1: $(\mathcal{C}, \mathcal{K}) \leftarrow \text{LLM}_\theta (\mathcal{P}_{\text{extract}}, R)$
2: $\{\mathbf{s}^{(1)}, \ldots, \mathbf{s}^{(n)}\} \leftarrow \text{UniProtAPI}(\mathcal{K})$
3: **for** each $\mathbf{s}^{(i)}, i = 1, \ldots, n$ **do**
4: $\quad \mathcal{R}_{\text{cons}}^{(i)} \leftarrow \text{ConservationAnalysis}(\mathbf{s}^{(i)}, \theta_{\text{cons}})$
5: $\quad$ Initialize $\mathcal{H}_0 \leftarrow \emptyset$
6: $\quad$ Initialize $\mathcal{S}_0 \leftarrow (\mathbf{s}^{(i)}, \xi_0, 0, \mathcal{H}_0)$
7: $\quad$ **for** $t = 1$ to $T_{\max}$ **do**
8: $\qquad \mathcal{D}_t \leftarrow \text{LLM}_\theta (\mathcal{P}_{\text{diag}}, \mathcal{S}_{t-1}, \mathcal{C}, \mathcal{H}_{t-1})$ *// THINK*
9: $\qquad \mathcal{Q}_t \leftarrow \text{LLM}_\theta (\mathcal{P}_{\text{plan}}, \mathcal{D}_t, \mathcal{S}_{t-1}, \mathcal{C}, \mathcal{H}_{t-1})$ *// PLAN*
10: $\qquad \mathbf{s}_t \leftarrow \text{Execute}\left(\mathbf{s}_{t-1}, \mathcal{Q}_t, \mathcal{R}_{\text{cons}}^{(i)}, \theta_{\text{ddG}}\right)$ *// ACT*
11: $\qquad \xi_t \leftarrow \text{Evaluate}(\mathbf{s}_t, \mathcal{C})$ *// OBSERVE*
12: $\qquad \mathcal{H}_t \leftarrow \mathcal{H}_{t-1} \cup \{(\mathcal{S}_{t-1}, \mathcal{Q}_t, \mathbf{s}_t, \xi_t)\}$
13: $\qquad$ Update $\mathcal{S}_t \leftarrow (\mathbf{s}_t, \xi_t, t, \mathcal{H}_t)$
14: $\qquad b_{\text{cont}} \leftarrow \text{LLM}_\theta (\mathcal{P}_{\text{reflect}}, \mathcal{S}_{t-1}, \mathcal{S}_t)$ *// REFLECT*
15: $\qquad$ **if** $\xi_t = 1.0$ **or** $\neg b_{\text{cont}}$ **then**
16: $\qquad\quad$ **break**
17: $\qquad$ **end if**
18: $\quad$ **end for**
19: **end for**
20: $\mathbf{s}^* \leftarrow \arg\max_{\mathbf{s}^{(i)}} \xi_{\text{final}}^{(i)}$
21: **return** $\mathbf{s}^*$

---

and `salt bridges` (charged residue pair requirements).

## 3.2. System Architecture Overview

Our PDAgent addresses this optimization problem through multiple phases illustrated in Figure 1. The process begins with an LLM-powered requirement analysis that extracts structured constraints and keywords from the natural language request. These keywords are used to retrieve template sequences that encode relevant functional motifs from the UniProt database (Ahmad et al., 2025; UniProt, 2025). Then, conservation analysis is adopted using BLAST (Camacho et al., 2009) and multiple sequence alignment (MSA) (Chatzou et al., 2016) to identify evolutionarily significant positions that should be protected from mutation. Inspired by the ReAct paradigm (Yao et al., 2022), we employ a five-phase reasoning loop for sequence refinement, including THINK, PLAN, ACT, OBSERVE, and REFLECT. Finally, comprehensive analysis evaluates the final sequence. The complete procedure is summarized in Algorithm 1.

## 3.3. Requirement Analysis

The requirement analysis phase transforms the natural language request $R$ into structured constraints and search key-words using an LLM with a specialized extraction prompt:

$$(\mathcal{C}, \mathcal{K}) = \text{LLM}_\theta (\mathcal{P}_{\text{extract}}, R) \tag{2}$$

where $\mathcal{P}_{\text{extract}}$ denotes the extraction prompt template (detailed in Appendix B.1), and $\mathcal{K} = \{k_1, k_2, \ldots, k_J\}$ is the search keywords set for template retrieval. The prompt instructs the LLM to extract numerical targets (temperature, pH, length), boolean requirements (solubility, disulfide bonds, salt bridges), structural preferences, and UniProt search keywords from the natural language request.

## 3.4. Candidates Preparation

The candidate generation phase queries the reviewed UniProt/Swiss-Prot database using extracted keywords $\mathcal{K}$. Retrieved sequences serve as templates for subsequent mutation-based optimization, providing reasonable starting points. Given keywords $\mathcal{K}$, we retrieve $n$ candidate template sequences (Appendix D):

$$\{\mathbf{s}^{(1)}, \mathbf{s}^{(2)}, \ldots, \mathbf{s}^{(n)}\} = \text{UniProtAPI}(\mathcal{K}) \tag{3}$$

To identify evolutionarily significant positions that should be protected during mutagenesis, we perform conservation analysis for each candidate sequence (Ashkenazy et al., 2016; Ingles-Prieto et al., 2013). This involves BLAST search against SwissProt to identify homologous sequences, MSA of top hits using MAFFT (Rozewicki et al., 2019), and position-specific conservation scoring using Shannon entropy (Lesne, 2014). For each candidate $\mathbf{s}^{(i)}$, positions with conservation scores exceeding threshold $\theta_{\text{cons}}$ are marked as highly conserved ($\mathcal{R}_{\text{cons}}^{(i)}$) and protected from mutation, while variable positions ($\mathcal{R}_{\text{var}}^{(i)}$) are candidates for optimization.

## 3.5. ReAct Optimization Loop

The core of our PDAgent framework is the ReAct loop that iteratively refines the protein sequence through five phases: THINK, PLAN, ACT, OBSERVE, and REFLECT.

**State Representation.** The optimization state at iteration $t$ is defined as:

$$\mathcal{S}_t = (\mathbf{s}_t, \xi_t, t, \mathcal{H}_t) \tag{4}$$

where $\mathbf{s}_t \in \mathcal{A}^L$ is the current sequence, $\xi_t \in [0, 1]$ is the constraint satisfaction rate, $t$ is the iteration index, and $\mathcal{H}_t = \{(\mathcal{S}_0, \mathcal{Q}_1, \mathbf{s}_1, \xi_1), \ldots, (\mathcal{S}_{t-1}, \mathcal{Q}_t, \mathbf{s}_t, \xi_t)\}$ is the optimization history. The history accumulates records from all previous iterations.

**THINK Phase.** The THINK phase employs the LLM to analyze the current state and identify optimization priorities:

$$\mathcal{D}_t = \text{LLM}_\theta (\mathcal{P}_{\text{diag}}, \mathcal{S}_{t-1}, \mathcal{C}, \mathcal{H}_{t-1}) \tag{5}$$

where $\mathcal{P}_{\text{diag}}$ is the diagnosis prompt (Appendix B.2), the output $\mathcal{D}_t = (\mathcal{I}_t, \mathcal{U}_t, \mathcal{O}_t)$ contains identified issues $\mathcal{I}_t$, hypothesized underlying causes $\mathcal{U}_t$, and priority ordering $\mathcal{O}_t$ for optimization. The history context enables the LLM to recognize patterns from previous iterations and avoid repeating ineffective strategies.

**PLAN Phase.** Based on the diagnosis $\mathcal{D}_t$, the LLM formulates an optimization strategy by selecting from a toolkit of seven mutation operators $\mathcal{F} = \{f_{\text{therm}}, f_{\text{sol}}, f_{\text{pI}}, f_{\text{SS}}, f_{\text{salt}}, f_{\text{helix}}, f_{\text{core}}\}$:

$$\mathcal{Q}_t = \text{LLM}_\theta(\mathcal{P}_{\text{plan}}, \mathcal{D}_t, \mathcal{S}_{t-1}, \mathcal{C}, \mathcal{H}_{t-1}) \qquad (6)$$

where $\mathcal{P}_{\text{plan}}$ is the planning prompt template (Appendix B.3). The action plan $\mathcal{Q}_t = (g_t, \mathbf{a}_t, \mathcal{E}_t, \mathcal{W}_t)$ specifies a goal $g_t$, an ordered sequence of $m$ actions $\mathbf{a}_t$ (each consisting of a tool from $\mathcal{F}$), expected outcomes $\mathcal{E}_t$, and potential risks $\mathcal{W}_t$.

Our toolkit $\mathcal{F}$ includes (detailed in Appendix C): $f_{\text{therm}}$ replaces thermolabile residues with stabilizing alternatives; $f_{\text{sol}}$ introduces polar/charged residues at surface-exposed hydrophobic patches; $f_{\text{pI}}$ adjusts charged residue balance; $f_{\text{SS}}$ introduces cysteine pairs for disulfide bonds (Craig & Dombkowski, 2013); $f_{\text{salt}}$ positions complementary charged residues for salt bridges (Bosshard et al., 2004); $f_{\text{helix}}$ promotes $\alpha$-helix formation (Azzarito et al., 2013); $f_{\text{core}}$ optimizes hydrophobic core packing (Kalinowska et al., 2017).

**ACT Phase.** The execution phase takes the action plan $\mathcal{Q}_t$ and applies each mutation sequentially:

$$\mathbf{s}_t = \text{Execute}(\mathbf{s}_{t-1}, \mathcal{Q}_t, \mathcal{R}_{\text{cons}}, \theta_{\text{ddG}}) \qquad (7)$$

Each proposed mutation specifying position $p$ is validated when $p \notin \mathcal{R}_{\text{cons}}$ and $\Delta\Delta G < \theta_{\text{ddG}}$, where $\Delta\Delta G$ is predicted using DDGun (Montanucci et al., 2019) and threshold $\theta_{\text{ddG}}$ prevents severely destabilizing mutations.

**OBSERVE Phase.** The observation phase re-analyzes the mutated sequence across all constraint dimensions. The foldability is assessed using ESMFold (Lin et al., 2023) with mean pLDDT (Jumper et al., 2021; Abramson et al., 2024) as the primary metric. We compute the updated constraint satisfaction rate $\xi_t$ and track changes for each individual constraints.

**REFLECT Phase.** The LLM analyzes iteration effectiveness and whether to continue optimization. It computes constraint-level changes and categorizes constraints as improved, worsened, or unchanged:

$$(b_{\text{cont}}, \mathcal{I}_{\text{insight}}, \tau_{\text{next}}) = \text{LLM}_\theta(\mathcal{P}_{\text{reflect}}, \mathcal{S}_{t-1}, \mathcal{S}_t, \mathcal{H}_t) \quad (8)$$

where $\mathcal{P}_{\text{reflect}}$ is the reflection prompt (Appendix B.4), $b_{\text{cont}}$ is the boolean continuation decision, $\mathcal{I}_{\text{insight}}$ is the insights about effective strategies, and $\tau_{\text{next}}$ recommended focus for the next iteration.

## 4. Experiments

### 4.1. Experimental Setup

**Benchmark Dataset.** Existing protein design benchmarks primarily focus on structure-conditioned design or rely on predefined functional annotations, which do not align with our natural language-driven paradigm. To address this gap, we curated a benchmark of 100 protein design tasks using Claude-Opus-4.5 (Anthropic, 2025b) representing diverse real-world design scenarios. Each task specifies 2-5 constraints with realistic parameter ranges. The distribution includes: `thermostability` (80%), `solubility` (68%), `pH stability` (15%), `length constraints` (29%), `structure preferences` (33%), `disulfide bonds` (11%), and `salt bridges` (9%). The complete dataset is provided in Appendix G.

**Evaluation Metrics.** We evaluate methods using three primary metrics. **Success Rate (SR)** measures the percentage of tasks producing valid protein sequences. **Foldability** reports the mean pLDDT score from ESMFold (Lin et al., 2023) predictions, indicating structural confidence with higher values suggesting more reliable protein folds. **Constraint Satisfaction Rate (CSR)** measures the percentage of constraints satisfied relative to the request specifications, computed separately for each constraint type and averaged across all constraints.

**Baseline Methods.** We compare against two categories of baselines. The first includes direct LLM generation using frontier models: GPT-4o (OpenAI, 2024), GPT-5.2 (OpenAI, 2025a), o4-mini (OpenAI, 2025b), Gemini-3-Pro (DeepMind, 2025), Claude-Sonnet-4.5 (Anthropic, 2025c), Claude-Haiku-4.5 (Anthropic, 2025a), DeepSeek-V3.2 (DeepSeek-AI, 2025), and Grok-4 (xAI, 2025), which receive the design request and generate sequences directly without iterative optimization (prompt provided in Appendix B.5). The second includes specialized deep learning methods for text-conditioned protein generation: ProtDAT (Guo et al., 2025), ProteinDT (Liu et al., 2025), Pinal (Dai et al., 2024), BioMedGPT-10B (Zhang et al., 2024; Peng et al., 2025), and Biomni (Huang et al., 2025).

**Implementation Details.** We evaluate PDAgent using three LLM backbones: LLaMA-3.1-8B (Llama Team, 2024), Qwen-3-30B (Qwen Team, 2025), and DeepSeek-V3.2 (DeepSeek-AI, 2025), with temperature set to 0.3 for diagnosis and reflection phases and 0.5 for planning. Structure prediction employs ESMFold with default parameters. Conservation analysis uses local BLAST against SwissProt with an E-value threshold of 0.001, identity threshold of 30%, and coverage threshold of 50%. We set the maximum number of iterations to 10, conservation threshold $\theta_{\text{cons}} = 0.6$, and stability threshold $\theta_{\text{ddG}} = 2.0$ kcal/mol. For each it-

*Table 1.* Performance comparison across different models and metrics. SR values represent success rates (%). Fold. (Foldability) is mean pLDDT. CSR values represent constraint satisfaction rates (%) for each dimension. Best results in **bold**, second best underlined, third best under-waved.

| Model | SR (%) | Fold. | CSR (%) | | | | | | | |
|---|---|---|---|---|---|---|---|---|---|---|
| | | | Thermo. | Solu. | pH | Length | Struct. | Disulf. | Salt | Avg. |
| *LLM Directly Prompting Methods* | | | | | | | | | | |
| GPT-4o (OpenAI, 2024) | 96 | 37.62 | 29.87 | 78.46 | 77.78 | 81.48 | 81.25 | 90.00 | **100.00** | 76.98 |
| GPT-5.2 (OpenAI, 2025a) | 93 | 47.84 | 28.77 | 82.81 | 73.68 | 76.92 | 96.67 | 87.50 | **100.00** | 78.05 |
| o4-mini (OpenAI, 2025b) | 62 | 27.50 | 22.45 | 43.18 | 63.64 | **100.00** | 90.48 | **100.00** | **100.00** | 74.25 |
| Gemini-3-Pro (DeepMind, 2025) | 95 | 50.03 | **60.53** | 61.54 | 58.82 | **100.00** | 96.67 | **100.00** | **100.00** | 82.55 |
| Claude-Sonnet-4.5 (Anthropic, 2025c) | 76 | 30.52 | 26.23 | **100.00** | 88.24 | 30.00 | 82.14 | **100.00** | **100.00** | 75.23 |
| Claude-Haiku-4.5 (Anthropic, 2025a) | 98 | 42.99 | 16.67 | 18.18 | 42.11 | 10.34 | 81.82 | 63.64 | **100.00** | 47.54 |
| DeepSeek-V3.2 (DeepSeek-AI, 2025) | 100 | 52.86 | 20.00 | 72.05 | 36.84 | 3.45 | 51.52 | 0.00 | 77.78 | 37.38 |
| Grok-4 (xAI, 2025) | 99 | 45.59 | 37.97 | 89.55 | 47.37 | 51.72 | 84.38 | 54.55 | **100.00** | 66.51 |
| *Other Protein Design Methods* | | | | | | | | | | |
| ProtDAT (Guo et al., 2025) | 100 | 46.91 | 3.75 | 66.18 | 53.33 | 3.45 | 39.39 | 27.27 | **100.00** | 41.91 |
| ProteinDT (Liu et al., 2025) | 100 | 39.15 | 2.50 | 66.18 | 40.00 | 10.34 | 60.61 | 27.27 | 88.89 | 42.26 |
| Pinal (Dai et al., 2024) | 100 | 80.92 | 14.29 | 64.71 | 53.33 | 13.79 | 63.64 | 63.64 | **100.00** | 53.34 |
| BioMedGPT-10B (Zhang et al., 2024) | 85 | 72.88 | 25.76 | **100.00** | 0.00 | 26.92 | 48.15 | 0.00 | 0.00 | 28.69 |
| Biomni (Huang et al., 2025) | 50 | 35.08 | 5.26 | 88.57 | 41.67 | 40.00 | 55.56 | 57.14 | **100.00** | 55.46 |
| *Ours* | | | | | | | | | | |
| PDAgent (LLaMA-3.1-8B) | 100 | **87.69** | 46.25 | **100.00** | 93.99 | **100.00** | 93.94 | **100.00** | **100.00** | 90.60 |
| PDAgent (Qwen-3-30B) | 100 | 80.54 | 55.00 | **100.00** | 80.00 | **100.00** | 87.88 | 72.73 | **100.00** | 85.09 |
| PDAgent (DeepSeek-V3.2) | 100 | **87.69** | 46.15 | **100.00** | **100.00** | **100.00** | 96.88 | **100.00** | **100.00** | **91.86** |

*Table 2.* Ablation study on framework components with backbone LLaMA-3.1-8B. The Average column reports the mean value of Foldability and CSR.

| Configuration | Fold. | CSR (%) | Average |
|---|---|---|---|
| **PDAgent** | **87.69** | **90.60** | **89.15** |
| − History Memory | 87.49 | 89.36 | ↓ 0.72 |
| − Iteration Loop | 87.80 | 81.85 | ↓ 4.32 |
| − THINK/PLAN Sep. | 87.83 | 86.53 | ↓ 1.97 |
| − Conservation Guard | 88.33 | 89.65 | ↓ 0.16 |
| Template Only | 88.68 | 82.24 | ↓ 3.69 |
| Random Generation | 30.24 | 55.46 | ↓ 46.3 |

eration, the per-tool mutation limit is 5 and the maximum number of actions is 4. All experiments are conducted on a single NVIDIA RTX 3090 GPU.

## 4.2. Main Results

Table 1 presents comprehensive results across all 100 design tasks. PDAgent (with DeepSeek-V3.2 backbone) achieves a remarkable 91.86% average CSR and 87.69 foldability, establishing state-of-the-art performance. PDAgent achieves 100% SR across all configurations, generating valid sequences for every task. In contrast, several baselines fail to produce valid sequences for substantial fractions of tasks: o4-mini succeeds on only 62% of tasks, BioMedGPT-10B

on 85%, and Biomni on merely 50%. We also report micro-average CSR to account for class imbalance: PDAgent (DeepSeek-V3.2) achieves 82.08%, outperforming Gemini-3-Pro (74.15%) and Pinal (43.70%). Relative rankings remain consistent with macro-average results.

The results also reveal an important trade-off between structural quality and constraint satisfaction. Specialized protein design models such as Pinal and BioMedGPT-10B achieve reasonable foldability but struggle significantly with complex multi-constraint satisfaction (CSR $< 54\%$), indicating their limitation in aligning semantic instructions with functional property optimization. Conversely, although some methods like Gemini-3-Pro and GPT-5.2 achieve relatively high average CSR, their foldability scores are only around 50, suggesting poor structural quality. In contrast, our PDAgent achieves the best balance between foldability and constraint satisfaction. We further note that the relatively low Thermo. CSR across all methods reflects the difficulty of the targets themselves: 52.5% of tasks require $T_m \geq 70°C$ and 21.25% require $T_m \geq 90°C$, values challenging to reach via point mutations alone.

## 4.3. Ablation Studies

To understand the contribution of each component, we conducted ablation studies on the $n = 3$ candidate configuration. Table 2 presents results with PDAgent using LLaMA-3.1-8B backbone.

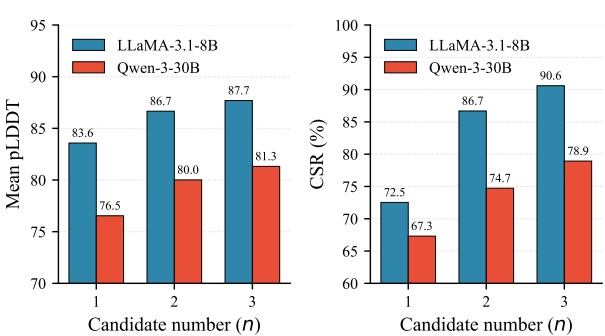

*Figure 2.* Performance comparison across LLM backbones and candidate numbers.

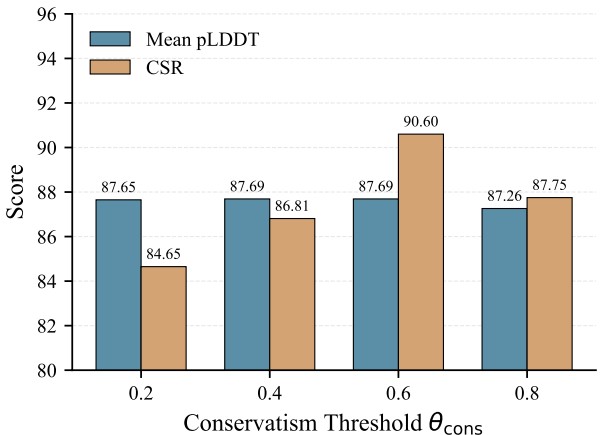

*Figure 3.* Effect of conservation threshold on design performance. Experiments conducted with LLaMA-3.1-8B and three candidate sequences.

Removing the iteration loop causes the largest average performance drop ($\downarrow$ 4.32), confirming that iterative refinement is essential for achieving complex design objectives. Removing the history memory reduces average performance by 0.72, demonstrating that accumulated insights from prior optimization attempts guide more effective subsequent decisions. Merging THINK and PLAN phases decreases performance by 1.97, indicating that explicit separation between diagnosis and planning improves optimization quality. Removing the conservation guard produces a minimal decrease.

Interestingly, the last three ablations slightly *increase* pLDDT but reduce CSR. This reveals an important insight: higher pLDDT does not necessarily indicate better protein design outcomes. The original UniProt templates already possess reasonable structural quality (Template Only reaches 82.24% CSR), and mutations required to satisfy user-specified constraints may marginally reduce structural confidence. For instance, removing the conservation guard allows mutations at conserved but structurally flexible po-

*Table 3.* ESMFold vs. AlphaFold3 structural quality on 30 sampled designs. The LLM backbone is DeepSeek-V3.2.

| Model | ESMFold Fold. | AF3 Fold. | Pearson $r$ |
|---|---|---|---|
| PDAgent | 87.85 | 91.00 | 0.92 |
| Pinal | 86.91 | 87.23 | 0.88 |
| Gemini-3-Pro | 52.32 | 56.60 | 0.91 |

sitions, which can appear favorable to ESMFold while potentially impairing biological function. This trade-off confirms that our framework actively modifies sequences to meet design requirements rather than trivially returning template sequences. The random generation baseline shows dramatic performance collapse, demonstrating that structured, knowledge-guided mutations are substantially more effective than random exploration.

Additionally, we tested the impact of candidate template numbers using two LLM backbones (LLaMA-3.1-8B and Qwen-3-30B), as shown in Figure 2. Both mean pLDDT and CSR improve with increasing candidate count. This aligns with intuition: more diverse starting points increase the likelihood of finding suitable templates for optimization.

### 4.4. Conservation Analysis

We investigate the effect of the conservation threshold $\theta_{\text{cons}}$, which controls mutable positions derived from MSA. Figure 3 presents experiments with LLaMA-3.1-8B and $n = 3$ candidate sequences across thresholds ranging from 0.2 to 0.8. The results indicate that the conservation threshold exhibits a trade-off between structural preservation and optimization flexibility. A threshold of 0.2 overly restricts the mutation search space by protecting even moderately variable positions, limiting the system's ability to satisfy design constraints. Conversely, a threshold of 0.8 provides insufficient protection, allowing mutations at functionally important sites that lead to decreased structural quality. The optimal threshold of 0.6 balances these factors, protecting evolutionarily constrained positions while maintaining sufficient degrees of freedom for constraint optimization.

### 4.5. Cross-Validation with AlphaFold3

To verify that ESMFold-based foldability scores provide reliable comparative evaluation, we randomly sampled 30 designed sequences and re-evaluated them using AlphaFold3 (Abramson et al., 2024) via the AlphaFold Server in Table 3. The strong correlation confirms that ESMFold provides reliable comparative evaluation. We chose ESMFold for in-loop evaluation because AlphaFold3 requires approximately 180s per sequence (vs. ESMFold's 1.92s), making it too slow for real-time feedback during iterative optimization.

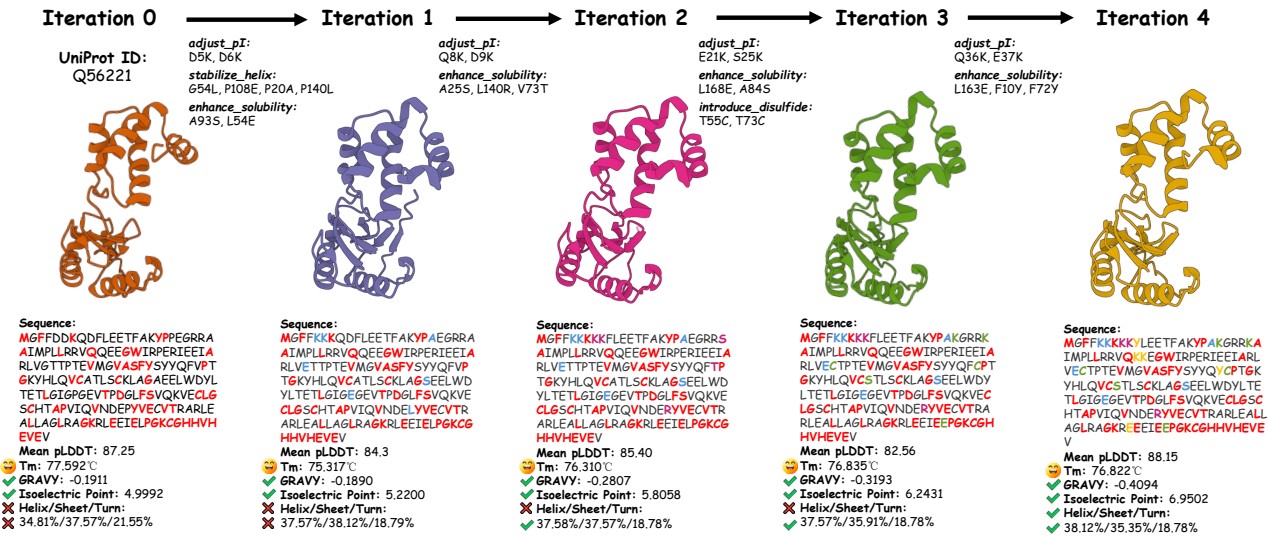

*Figure 4.* Optimization trajectory for the case study. pH stability is assessed via isoelectric point (pI) matching. The red residues are the highly conserved positions identified by MSA.

*Table 4.* Computational cost breakdown average per operation per design task (average with one candidate sequence).

| Component | Primary Operation | Time (s) |
|---|---|---|
| Candidate Generation | UniProt API Query | 7.07 |
| Conservation Analysis | BLAST + MSA[1] | 1.01 |
| Property Calculation | BioPython + DSSP | 0.25 |
| Stability Prediction | $T_m$ Estimation[2] | 13.32 |
| Structure Prediction | ESMFold Inference | 1.92 |
| LLM Reasoning | LLM Inference[3] | 3.36 |
| Mutation Proposal | $\Delta\Delta G$ Prediction[4] | 29.55 |
| **Total per Iteration** | — | **44.99** |
| **Total per Design** | (avg. 3.1 iterations) | **~139** |

[1] Local BLAST against SwissProt database with MAFFT for MSA.
[2] PPTStab for melting temperature ($T_m$) prediction.
[3] LLaMA-3.1-8B for LLM Inference.
[4] DDGun for mutation stability effect ($\Delta\Delta G$) prediction.

### 4.6. Computational Efficiency

Table 4 breaks down computational costs per iteration per design task with LLaMA-3.1-8B backbone and one candidate sequence. Experiments were conducted on a single NVIDIA RTX 3090 GPU and Intel Xeon Gold 6240C CPU. The most computationally intensive operations are mutation proposal (29.55s) and stability prediction (13.32s), as they require deep learning inference with DDGun (Montanucci et al., 2019) and PPTStab (Tijare et al., 2025). With an average of 3.1 iterations per design task, complete optimization requires approximately 2.3 minutes. This efficiency demonstrates that PDAgent is a practical and scalable solution for

high-throughput *in silico* protein design.

## 5. Case Study

To illustrate PDAgent's optimization process, we present a detailed case study for the following design specification: *"I need to engineer a lipase enzyme for diagnostic assay development with high thermostability to withstand 65°C with enhanced solubility in aqueous solutions engineered for alkaline environments at pH 8.5 rich in alpha-helical content."* (Task 16 in Table 12) Figure 4 visualizes the progressive improvement with 3D structure.

The system extracts four constraints: thermostability ($T_m \geq$ 65°C), solubility (GRAVY < 0), pH stability (pI $\approx$ 8.5), and structure preference ($\alpha$-helix rich). From UniProt search, the agent selects Q56221, a thermostable lipase with mean pLDDT of 87.25. Conservation analysis identified 29 homologs and recognized highly conserved positions. Initial analysis shows 2 of 4 constraints satisfied (thermostability and solubility).

Over four iterations, the system applies targeted mutations: **Iteration 1** applies adjust_pI, stabilize_helix, and enhance_solubility. **Iteration 2** applies adjust_pI and enhance_solubility, improving CSR from 50% to 75%. **Iteration 3** introduces additional refinements including a disulfide bond. **Iteration 4** applies final optimizations and achieves 100% CSR, yielding pLDDT of 88.15. The final optimized sequence satisfies all four design constraints with mutations distributed across non-conserved positions, demonstrating PDAgent's ability to systematically address multiple competing constraints

through iterative refinement while preserving the evolutionarily constrained core structure.

This behavior holds in aggregate: across all benchmark, refined sequences retain a mean identity of 92.81% (std 6.17) to their templates, confirming that PDAgent performs targeted, localized edits rather than substantive redesign. We also provide an additional thermostability-focused trajectory in Appendix F.

# 6. Discussion

Our PDAgent demonstrates that LLM-driven autonomous agents can effectively bridge the semantic gap between natural language protein design specifications and sequence-level optimization. The template-based approach offers distinct advantages over *de novo* generation (Watson et al., 2023; Ferruz et al., 2022): templates from curated databases provide structurally validated starting points, and conservation analysis identifies positions that can be safely modified without disrupting functional integrity. The iterative ReAct loop (Yao et al., 2022) enables adaptive refinement that single-pass generation methods cannot achieve, allowing the agent to diagnose constraint violations, apply targeted mutations, and learn from optimization history within each design task. More broadly, this suggests a useful pattern for agentic scientific design: pair closed-loop validation with domain-aware constraint guards. Unlike expert-driven tools such as Rosetta (Leaver-Fay et al., 2011; Leman et al., 2020), PDAgent substantially lowers the barrier for non-expert researchers to engage in protein design.

**Limitations.** Despite strong performance, PDAgent has several limitations. First, the current framework relies on computational predictors rather than wet-lab experimental validation, and prediction accuracy may vary across protein families and degrade for sequences far from each predictor's training distribution. We therefore position PDAgent as a high-throughput in silico screening step that precedes wet-lab validation. Second, our constraint taxonomy does not yet cover complex functional properties such as binding affinity or catalytic activity. Third, the framework's practical utility has so far been validated only in silico, leaving downstream applications and direct comparison against expert-driven engineering workflows to be explored. Future work will focus on extending the constraint taxonomy and integrating experimental feedback for closed-loop laboratory optimization, and on fostering standardized benchmarks to systematically evaluate text-guided protein design.

# 7. Conclusion

We presented PDAgent, an LLM-driven autonomous agent framework for *in silico* protein design via directed mutation. By combining natural language understanding with template retrieval, conservation-aware mutation strategies, and iterative ReAct-style optimization, PDAgent enables non-expert users to design proteins satisfying complex multi-constraint specifications. Experiments on 100 diverse design tasks demonstrate that PDAgent achieves 91.86% average constraint satisfaction rate with high structural quality (mean pLDDT 87.69), substantially outperforming both direct LLM generation methods and specialized deep learning methods. The framework's architecture facilitates extension to new constraint types and optimization tools, providing a foundation for LLM-assisted protein engineering workflows.

# Acknowledgement

This work is supported by the Fundamental and Interdisciplinary Disciplines Breakthrough Plan of the Ministry of Education of China (Grant No. JYB2025XDXM704), the New Cornerstone Science Foundation through the XPLORER PRIZE, the National Natural Science Foundation of China (Grant No. U23A20318, 62272354 and 62276195), the Foundation for Innovative Research Groups of Hubei Province (Grant No. 2024AFA017) and the Science and Technology Major Project of Hubei Province (Grant No. 2025BCB026). This work was also supported by WHU-Kingsoft Joint Lab. The numerical calculations in this paper have been done on the supercomputing system in the Supercomputing Center of Wuhan University.

# Impact Statement

This work aims to advance computational protein design by making it more accessible to researchers across disciplines. While protein engineering technologies carry dual-use considerations and could potentially be misused, we believe the benefits to biomedical research and biotechnology substantially outweigh the risks. We will continue to develop responsible usage guidelines as the framework evolves.

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

## A. Constraint Evaluation

This appendix provides the formal definitions of constraint satisfaction functions used in PDAgent.

### A.1. Satisfaction Function Definition

Given a constraint $c = (\tau, v, \gamma, r_{\min}, r_{\max})$, the binary satisfaction function $\sigma(\mathbf{s}, c)$ is defined according to the comparison operator $\gamma$:

$$\sigma(\mathbf{s}, c) = \begin{cases} \mathbb{I}[\psi_\tau(\mathbf{s}) \geq v] & \text{if } \gamma = \texttt{geq} \\ \mathbb{I}[|\psi_\tau(\mathbf{s}) - v| \leq \epsilon_\tau] & \text{if } \gamma = \texttt{eq} \\ \mathbb{I}[r_{\min} \leq \psi_\tau(\mathbf{s}) \leq r_{\max}] & \text{if } \gamma = \texttt{range} \\ \mathbb{I}[\psi_\tau(\mathbf{s}) \geq 1] & \text{if } \gamma = \texttt{bool} \end{cases} \qquad (9)$$

where $\psi_\tau(\mathbf{s})$ is the property evaluation function for constraint type $\tau$, $\mathbb{I}[\cdot]$ is the indicator function, and $\epsilon_\tau$ is a type-specific tolerance threshold.

### A.2. Constraint Taxonomy

Table 5 summarizes the constraint types, their associated property functions, and satisfaction criteria.

*Table 5.* Constraint taxonomy with property evaluation functions and satisfaction criteria.

| Type $\tau$ | Property $\psi_\tau$ | $\gamma$ | Satisfaction Criterion |
|---|---|---|---|
| Thermostability | Estimated $T_m$ (°C) | `geq` | $T_m^{\text{est}} \geq T_m^{\text{target}}$ |
| Solubility | GRAVY | `bool` | GRAVY $\leq 0$ |
| pH Stability | Isoelectric point $pI$ | `eq` | $\lvert pI - v \rvert \leq 2.0$ |
| Length | Sequence length $L$ | `range` | $r_{\min} \leq L \leq r_{\max}$ |
| Structure Pref. | Secondary structure fractions | `eq` | Type-specific[†] |
| Disulfide Bonds | Cys pair potential | `bool` | $n_{\text{SS}}^{\text{potential}} \geq 1$ |
| Salt Bridges | Charged pair potential | `bool` | $n_{\text{salt}}^{\text{potential}} \geq 1$ |

[†] Structure preference criteria: $\alpha$-helix dominant requires helix fraction $f_\alpha \geq 0.25$ and $f_\alpha > f_\beta$; $\beta$-sheet dominant requires sheet fraction $f_\beta \geq 0.20$ and $f_\beta > f_\alpha$; mixed structure requires both $f_\alpha \geq 0.15$ and $f_\beta \geq 0.15$.

## B. Prompt Templates

We provide the complete prompt templates used by LLM-powered components in PDAgent. These prompts are carefully designed to elicit structured JSON outputs that can be reliably parsed and executed by subsequent system components.

### B.1. Constraints Extraction

**Prompt**

```
You are a protein engineering expert. Analyze the following protein design request
    and extract structured information.

Design Request: {request}

Extract the following information and respond in JSON format ONLY (no other text):
{{
    "function": "Brief description of protein function",
    "length_min": <minimum length as integer, default 50>,
    "length_max": <maximum length as integer, default 200>,
    "temperature": <target working temperature in Celsius, null if not specified,
    use median if a range>,
    "pH": <target pH, null if not specified, use median if a range>,
```

```
        "thermostable": <true if thermostability is required, false otherwise>,
        "high_solubility": <true if high solubility is required, false otherwise>,
        "structure_preference": "<alpha_helix/beta_sheet/mixed/any>",
        "required_features": ["list of required features: disulfide, salt_bridge"],
        "keywords": ["UniProt search keywords for finding template sequences (maximum 4
    terms)"]
}}

Important:
1. keywords should be specific terms for searching UniProt database (e.g.,
    "thermophilic", "lipase", "alpha-amylase"), and
2. Include organism names if mentioned (e.g., "Thermus thermophilus")
3. If the request mentions specific protein families or enzymes, include them as
    keywords
4. **keywords must be limited to a MAXIMUM of 4 terms**
5. Prioritize selecting keywords in this order to ensure search relevance:
    - a. Explicitly mentioned protein families, enzyme classes, or specific protein
    names
    - b. Core terms describing the protein's primary biological function
    - c. Explicitly mentioned organism species names
    - d. Key functional/structural properties critical to the design
6. If the user wantted protein need to live at high temperature (such as 60 or
    higher), you can consider to include some heavly related keywords or species
    like hyperthermophile, pyrococcus", "thermococcus", "thermus", "thermotoga",
    "thermophile", "geobacillus", "thermobifida
7. If fewer than 4 relevant terms exist, include only the available valid terms; if
    no relevant terms, use an empty list
8. For required_features, only include "disulfide" or "salt_bridge" if explicitly
    mentioned in the request
```

## B.2. Diagnosis Prompt (THINK Phase)

**Prompt**

```
You are a protein engineering expert. Analyze the current protein state and
    identify which KEY CONSTRAINTS are NOT satisfied.

{context}

**Design Constraints**:
{constraints_desc}

{unsatisfied_desc}

Focus ONLY on these 7 evaluation dimensions:
1. thermostability - Is estimated Tm meeting the target?
2. solubility - Is GRAVY low enough (< 0)? Is instability index low (< 40)?
3. ph_stability - Is pI appropriate for the target pH?
4. length - Is length within the required range?
5. structure_preference - Does helix/sheet ratio match the target structure?
6. disulfide_bonds - Are there enough Cys pairs if required?
7. salt_bridges - Are there enough K/R-D/E pairs if required?

Identify issues ONLY related to these constraints. Prioritize unsatisfied
    constraints.
Consider the optimization history to avoid repeating ineffective strategies.

Respond in JSON format ONLY:
{{
    "issues": [
        {{
```

```
            "issue": "Brief description",
            "severity": "high/medium/low",
            "constraint_type": "one of the 7 types",
            "cause": "root cause"
        }}
    ],
    "root_causes": ["cause1", "cause2"],
    "priority_order": ["constraint_type1", "constraint_type2"]
}}

Focus on actionable issues that can be improved through mutation strategies.
```

### B.3. Planning Prompt (PLAN Phase)

**Prompt**

```
You are a protein engineer expert. Your task is to Analyze the current protein
    state and plan optimization actions to satisfy the KEY CONSTRAINTS.

{context}

**Unsatisfied Constraints** (in priority order):
{issues_desc}

**Priority Constraints to Address**: {', '.join(diagnosis.priority_order)}

**Available Tools**:
{tools_desc}

**Tool-Constraint Mapping**:
- thermostability -> enhance_thermostability, introduce_disulfide
- solubility -> enhance_solubility
- ph_stability -> adjust_pI
- structure_preference (alpha_helix) -> stabilize_helix
- disulfide_bonds -> introduce_disulfide
- salt_bridges -> introduce_salt_bridge

Plan 2-4 actions to address the MOST CRITICAL unsatisfied constraints. Select tools
    that directly target the constraint types identified.
Learn from the optimization history: prefer tools that have been effective, avoid
    repeating failed strategies.

Respond in JSON format ONLY:
{{
    "goal": "Main constraint(s) to satisfy in this iteration",
    "actions": [
        {{
            "tool": "tool_name from available tools",
            "parameters": {{"max_mutations": 3}},
            "rationale": "How this addresses the constraint",
            "target_constraint": "which constraint this targets"
        }}
    ],
    "expected_outcomes": ["outcome1", "outcome2"],
    "risks": ["potential risk1"]
}}
```

## B.4. Reflection Prompt (REFLECT Phase)

**Prompt**

```
You are a protein engineering expert. I did an optimization iteration and have the
    before-and-after analysis reports. Analyze the state change and provide insights.

# Before Optimization:
{old_context}

# After Optimization:
{new_context}

**Satisfaction Rate Change**: {old_rate:.1%} -> {new_rate:.1%} (\Delta = {(new_rate-
    old_rate)*100:+.1f}%)

**Constraint Changes**:
- Improved: {improved if improved else 'None'}
- Worsened: {worsened if worsened else 'None'}
- Still unsatisfied: {list(unsatisfied.keys()) if unsatisfied else 'None'}

Based on validation results and optimization history:
1. Which constraints improved/worsened?
2. What should be the priority for next iteration?
3. Should we continue or stop? (Consider if we're making progress or stuck in a loop
    )

Respond in JSON:
{{
    "should_continue": true/false,
    "insights": ["..."],
    "next_focus": "constraint_type",
    "effectiveness": "high/medium/low"
}}
```

## B.5. LLM Baseline Prompt

**Prompt**

```
You are an expert protein designer. Your task is to design a novel protein sequence
    based on the given requirements.

## Requirements:
{request}

## Design Guidelines:
1. Generate a valid amino acid sequence using only the 20 standard amino acids
    (ACDEFGHIKLMNPQRSTVWY)
2. Consider the following properties based on requirements:
   - For thermostability: Include more Pro, Ile, Val, Leu, Ala; Avoid Gln, Asn,
   Met, Cys
   - For solubility: Include charged residues (Lys, Arg, Glu, Asp); Avoid
   hydrophobic patches
   - For stability: Consider disulfide bonds (Cys pairs), salt bridges (Lys/Arg
   with Asp/Glu)
3. Default sequence length: 50-200 amino acids unless specified otherwise

## Output Format:
```

```
Provide your response in the following XML format (starting with <sequence> and
    ending with </sequence>):

<sequence>
YOUR_PROTEIN_SEQUENCE_HERE
</sequence>

Now design the protein sequence:
```

## C. Mutation Tools Description

Table 6 provides detailed descriptions of the seven mutation tools available in the PDAgent toolkit.

*Table 6.* Mutation tools available in PDAgent with their target constraints and mutation strategies.

| Tool | Target Constraint | Strategy |
|------|-------------------|----------|
| enhance_thermostability ($f_{\text{therm}}$) | Thermostability | Replaces oxidation- and deamidation-prone residues with stable alternatives: Gln→Glu, Asn→Asp/Ser, Met→Leu/Ile, Cys→Ser/Ala, Trp→Phe (Bhanuramanand et al., 2014; Kim et al., 2001). |
| enhance_solubility ($f_{\text{sol}}$) | Solubility | Introduces polar/charged residues at hydrophobic patches identified via sliding window (size 8, ≥6 hydrophobic residues): Ile→Thr/Ser/Lys, Leu→Lys/Arg/Glu, Val→Thr/Ser, Phe→Tyr, Ala→Ser, Met→Lys/Arg (Lawrence et al., 2007; Der et al., 2013; Yagi et al., 2014). |
| adjust_pI ($f_{\text{pI}}$) | pH Stability | Adjusts isoelectric point by modifying charged residues. To increase pI: Asp/Glu→Lys/Arg. To decrease pI: Lys/Arg/His→Glu/Asp. Polar residues (Ser, Thr, Asn, Gln) may also be converted to charged residues (Trevino et al., 2007; Shaw et al., 2001). |
| introduce_disulfide ($f_{\text{SS}}$) | Disulfide Bonds | Identifies candidate Cys pairs based on sequence separation (10–50 residues). Prioritizes positions with existing Cys/Ser/Thr and penalizes Pro/Gly at candidate positions. Introduces two Cys mutations to form potential disulfide bond (Dani et al., 2003; Craig & Dombkowski, 2013; Liu et al., 2016). |
| introduce_salt_bridge ($f_{\text{salt}}$) | Salt Bridges | Introduces complementary charged pairs (Lys–Glu) at sequence separations of 3–7 residues, corresponding to intra-helical i, i+3/4 geometry. Prioritizes positions already containing charged or polar residues (Bosshard et al., 2004; Lee et al., 2014; Ban et al., 2019). |
| stabilize_helix ($f_{\text{helix}}$) | Structure Preference | Replaces helix-breaking residues (Pro, Gly) with helix-promoting residues (Ala, Glu, Leu, Met) at variable positions (Serrano et al., 1992; Blaber et al., 1993; Azzarito et al., 2013). |
| optimize_core ($f_{\text{core}}$) | Overall Stability | Introduces hydrophobic residues (Val, Ile, Leu, Met, Phe) at estimated core positions, identified as regions with high local hydrophobicity (≥4 hydrophobic residues in 7-residue window). Small residues→Val, polar residues→Leu, other residues→Ile (Zhu et al., 1993; Kalinowska et al., 2017). |

## D. Template Retrieval Strategy

When the full keyword query returns insufficient candidates, PDAgent progressively relaxes the query by trying all keyword subsets (ordered from largest to smallest). The procedure is given in Algorithm 2.

**Algorithm 2** Template Candidate Generation

**Require:** Generated keywords $\mathcal{K} = \{k_1, k_2, \ldots, k_J\}$, constraints $\mathcal{C}$, target count $n$
**Ensure:** Candidate list $S$ with $|S| = n$
    *// Generate all keyword subsets from largest to smallest*
 1: SubsetList $\leftarrow [\mathcal{K}]$
 2: **for** $m \leftarrow 1$ **to** $J - 1$ **do**
 3:   **for** each index combination $D \in \binom{[J]}{m}$ **do**
 4:     SubsetList.APPEND($\{k_i \mid i \notin D\}$)
 5:   **end for**
 6: **end for**
    *// Retrieve template candidates*
 7: $S \leftarrow []$
 8: **for** $K' \in$ SubsetList **do**
 9:   $S$.EXTEND(UniProtAPI($K', \mathcal{C}, n$))
10:   **if** $|S| \geq n$ **then**
11:     **break**
12:   **end if**
13: **end for**
14: **while** $|S| < n$ **do**
15:   $S$.APPEND(RandomGenerate($\mathcal{C}$))
16: **end while**
17: **return** $S[0 : n]$

# E. More Experimental Results

## E.1. Rule-based Controller Comparison

To isolate the contribution of LLM reasoning from the overall pipeline, we implemented a rule-based controller that uses the same retrieval, conservation analysis, and mutation toolkit, but replaces the LLM with a deterministic policy that selects mutation operators based solely on which constraints are currently violated. We further split the 100 tasks into 76 non-conflicting and 24 conflicting tasks (e.g., simultaneously requiring high thermostability and high solubility).

*Table 7.* CSR performance of different settings. The LLM backbone is DeepSeek-V3.2.

| Configuration | Avg. CSR (%) | Conflict CSR (%) | Non-conflict CSR (%) |
|---|---|---|---|
| **PDAgent** | **91.86** | **84.69** | **92.64** |
| − LLM (Rule-based) | 88.55 (↓ 3.31) | 71.77 (↓ 12.92) | 91.72 (↓ 0.92) |
| − Iteration | 81.85 (↓ 10.01) | 69.73 (↓ 14.96) | 84.70 (↓ 7.94) |

As shown in Table 7, both iteration and LLM reasoning contribute to performance. Removing iteration causes a 10.01-point drop, while the overall LLM-ablation gap is 3.31 points, but the LLM's contribution concentrates on conflicting tasks (+12.92 points), where fixed rules cannot dynamically balance competing objectives. This confirms that iteration is the key contributor, while LLM reasoning is essential for handling trade-offs in complex scenarios.

## E.2. Template Diversity vs. Multiple Stochastic Runs

To quantify how much of the performance gain comes from template diversity rather than multiple optimization runs, we ran a controlled comparison in Table 8: optimizing the same single template three times independently and selecting the best result.

*Table 8.* Performance with different template settings. The LLM backbone is LLaMA-3.1-8B.

| Setting | CSR (%) | Fold. |
|---|---|---|
| PDAgent, n=3 diverse templates (default) | 90.60 | 87.69 |
| Single template ×3 runs, oracle selection | 78.33 | 83.20 |
| Single template ×1 run | 74.51 | 83.06 |

Oracle selection across multiple stochastic runs on a single template adds only 3.82 CSR points, while template diversity adds 12.27 points. This confirms that the performance gain primarily comes from diverse starting points rather than multiple stochastic runs, validating our design choice of retrieving multiple templates.

### E.3. Novelty and Diversity Analysis

To examine the novelty-reliability trade-off, we compared PDAgent against baseline methods on sequence novelty and intra-set pairwise diversity (defined as $1 - \bar{p}$, where $\bar{p}$ is the mean pairwise sequence identity within design set):

*Table 9.* Novelty and diversity comparison across methods.

| Method | Mean identity to UniProt hit (%) | Intra-set pairwise diversity (%) |
|---|---|---|
| PDAgent | 92.81 | 75.28 |
| Pinal | 61.88 | 78.78 |
| Direct LLM (Gemini) | 10.99 | 70.82 |

As shown in Table 9, PDAgent's higher sequence identity to known UniProt entries is intentional: our template-based approach deliberately conserves known functional scaffolds to ensure structural reliability. Sequences generated directly by LLMs are highly novel but exhibit poor structural quality (mean pLDDT $< 53$), precisely because they lack the evolutionary grounding present in natural proteins.

### E.4. Backbone-wise Behavior Analysis

Beyond Table 1, we analyze per-backbone behavior on the 76 non-conflicting and 24 conflicting tasks:

*Table 10.* Performance with Different Backbone Models

| Backbone | Fold. | Avg. CSR (%) | Avg. #Mutations | Conflict CSR (%) | Non-conflict CSR (%) | Avg. #Iter |
|---|---|---|---|---|---|---|
| LLaMA-3.1-8B | 87.69 | 90.60 | 6.05 | 84.69 | 92.64 | 3.14 |
| Qwen-3-30B | 80.54 | 85.09 | 2.00 | 75.01 | 88.84 | 1.37 |
| DeepSeek-V3.2 | 87.69 | 91.86 | 13.02 | 89.12 | 92.93 | 10.41 |

The results in Table 10 reveal: (1) Qwen-3-30B's lower performance is mainly due to early stopping (1.37 iterations, 2.00 mutations per task on average), suggesting its REFLECT phase terminates optimization too aggressively; (2) LLaMA-3.1-8B reaches competitive performance with fewer iterations than DeepSeek-V3.2, showing that smaller models can work effectively within our framework; (3) the performance gap between backbones is larger on conflicting tasks, confirming that stronger LLM reasoning is most valuable when trade-offs are involved.

## F. Case Study: Thermostability Starting Unmet

To demonstrate PDAgent's ability to satisfy initially unmet thermostability constraints, we present the following task: *"Design a kinase for biofuel production applications with optimal activity around 75°C, engineered for alkaline environments at pH 6.5, with improved expression and solubility characteristics."* (Task 79 in Table 12)

The retrieved template (UniProt: B9K712) has $T_m = 73.64$°C, failing the $T_m \geq 75$°C requirement. Over four iterations,

PDAgent applies `enhance_thermostability` and `adjust_pI` operators, gradually increasing $T_m$ to 75.44°C and adjusting pI to 6.50, reaching 100% CSR at iteration 4. The full optimization trajectory is shown in Table 11.

*Table 11.* Optimization trajectory for appendix case.

| Iteration | $T_m$ (°C) | GRAVY | pI | CSR | Mutations |
|:---:|:---:|:---:|:---:|:---:|:---:|
| 0 | 73.64 | −0.60 | 9.50 | 33% | — |
| 1 | 74.19 | −0.61 | 9.08 | 33% | K9E, K10E, C64S, N74D |
| 2 | 74.20 | −0.63 | 8.56 | 33% | R16E, G58P, F143Y |
| 3 | 74.96 | −0.63 | 5.85 | 33% | K19E, Q59E |
| 4 | 75.44 | −0.66 | 6.50 | 100% | S30P, L56K, S64C |

## G. Benchmark Dataset

Table 12 provides the protein design tasks we used in our experiments. Tasks were generated using Claude-Opus-4.5 with prompts designed to ensure diverse coverage of all constraint categories.

*Table 12.* Benchmark dataset of 100 protein design tasks.

| ID | Requests |
|:---:|:---|
| 1 | I'm looking to develop a storage protein for textile industry applications with high thermostability to withstand 75°C with enhanced overall structural integrity designed for acidic conditions around pH 6.0 comprising 150-300 amino acids with predominantly alpha-helical secondary structure. |
| 2 | I require a hydrolase for agricultural applications that avoids aggregation and maintains solubility approximately 120 to 200 residues in length with excellent solubility properties for downstream processing. |
| 3 | I'm interested in creating a signaling protein for biosensor applications capable of maintaining activity at 90°C with improved expression and solubility characteristics engineered for alkaline environments at pH 5.5 rich in alpha-helical content. |
| 4 | I want to engineer an esterase for environmental remediation that can tolerate temperatures as high as 80°C that functions optimally at pH 8.0 The protein should maintain activity above 80°C. |
| 5 | I would like to engineer a cellulase for biofuel production applications approximately 120 to 200 residues in length that avoids aggregation and maintains solubility The protein needs to be compatible with standard purification methods. |
| 6 | I want to create an industrial enzyme for pharmaceutical manufacturing processes that remains stable and active at 70°C with enhanced solubility in aqueous solutions in the range of 80 to 180 residues. |
| 7 | I require a protease for agricultural applications that remains soluble at high concentrations The protein should maintain activity above 70°C. The design should avoid oxidation-sensitive residues where possible. |
| 8 | I need to engineer a phosphatase for biotechnology research that remains stable and active at 55°C with a mixed alpha-beta fold with enhanced solubility in aqueous solutions. |
| 9 | I'm looking to develop a storage protein for paper and pulp processing with improved expression and solubility characteristics High expression yield in bacterial systems is preferred. |
| 10 | I need to develop a protease for brewing and fermentation that remains stable and active at 70°C with excellent solubility properties for downstream processing with enhanced overall structural integrity. |
| 11 | I want to create a xylanase for waste treatment processes designed to operate efficiently at 65°C with enhanced solubility in aqueous solutions Long-term storage stability is important. |

*(continued from previous page)*

| ID | Requests |
| --- | --- |
| 12 | I require a reductase for agricultural applications that can function at temperatures up to 75°C incorporating multiple stabilizing features stable across a pH range of 7.5 to 10.5 comprising 150-230 amino acids The protein should have minimal aggregation tendency. |
| 13 | I'm interested in creating a peptidase for detergent formulations that can tolerate temperatures as high as 55°C with excellent solubility properties for downstream processing that functions optimally at pH 10.0. |
| 14 | I need to design a binding protein for detergent formulations that can function at temperatures up to 55°C with disulfide bonds for enhanced structural stability that functions optimally at pH 9.0. |
| 15 | I need to develop a regulatory protein for paper and pulp processing that avoids aggregation and maintains solubility incorporating multiple stabilizing features engineered for alkaline environments at pH 7.0 with a mixed alpha-beta fold. |
| 16 | I need to engineer a lipase enzyme for diagnostic assay development with high thermostability to withstand 65°C with enhanced solubility in aqueous solutions engineered for alkaline environments at pH 8.5 rich in alpha-helical content. |
| 17 | I want to create a cellulase for detergent formulations with optimal activity around 65°C with excellent solubility properties for downstream processing incorporating salt bridges for improved stability. |
| 18 | I'm looking to develop a transferase for diagnostic assay development that can tolerate temperatures as high as 50°C with enhanced solubility in aqueous solutions with stable beta-sheet regions The protein needs to be compatible with standard purification methods. |
| 19 | I want to create a cellulase for textile industry applications that avoids aggregation and maintains solubility with a length between 100 and 200 amino acids with predominantly alpha-helical secondary structure The design should avoid oxidation-sensitive residues where possible. |
| 20 | I'm looking to develop a kinase for pharmaceutical manufacturing processes with high thermostability to withstand 80°C Long-term storage stability is important. that remains soluble at high concentrations. |
| 21 | I want to engineer a hydrolase for brewing and fermentation designed to operate efficiently at 65°C with stable beta-sheet regions The protein should be amenable to further engineering. |
| 22 | I need to design a kinase for food processing industry use that remains stable and active at 65°C with excellent solubility properties for downstream processing stable across a pH range of 3.5 to 6.5 approximately 150 to 300 residues in length. |
| 23 | I need to design a peptidase for biotechnology research with high thermostability to withstand 55°C that avoids aggregation and maintains solubility incorporating salt bridges for improved stability stable across a pH range of 6.0 to 9.0. |
| 24 | I want to design a peroxidase for biosensor applications that remains stable and active at 65°C that avoids aggregation and maintains solubility with a mixed alpha-beta fold. |
| 25 | I'm interested in creating an isomerase for textile industry applications that remains soluble at high concentrations with enhanced overall structural integrity stable across a pH range of 3.0 to 5.5 approximately 150 to 230 residues in length with stable beta-sheet regions. |
| 26 | I need to develop a protease for textile industry applications that functions optimally at pH 7.0 The design should avoid oxidation-sensitive residues where possible. The protein should maintain activity above 65°C. |
| 27 | I would like to engineer a hydrolase for molecular biology applications that can function at temperatures up to 95°C with optimized hydrophobic core packing rich in alpha-helical content The protein should have minimal aggregation tendency. |
| 28 | I require a transporter protein for food processing industry use that avoids aggregation and maintains solubility It should maintain structural integrity under stress conditions. The protein should maintain activity above 80°C. |

*(continued on next page)*

*(continued from previous page)*

| ID | Requests |
| --- | --- |
| 29 | I want to engineer a storage protein for agricultural applications with excellent solubility properties for downstream processing Long-term storage stability is important. The design should avoid oxidation-sensitive residues where possible. |
| 30 | I need to develop a lyase for diagnostic assay development incorporating salt bridges for improved stability stable across a pH range of 3.5 to 6.5 in the range of 100 to 200 residues. |
| 31 | I'm interested in creating an antimicrobial peptide for therapeutic drug development featuring stabilized alpha-helical regions with broad pH tolerance from 6.5 to 9.5 The protein needs to be compatible with standard purification methods. |
| 32 | I'm looking to develop a hydrolase for agricultural applications with high thermostability to withstand 80°C incorporating multiple stabilizing features with a length between 80 and 230 amino acids The protein should have minimal aggregation tendency. |
| 33 | I need to engineer a xylanase for detergent formulations that can tolerate temperatures as high as 75°C that exhibits high solubility and minimal aggregation in the range of 120 to 270 residues Long-term storage stability is important. |
| 34 | I need to design a ligase for environmental remediation with high thermostability to withstand 55°C that avoids aggregation and maintains solubility The protein should maintain activity above 75°C. |
| 35 | I want to create a peptidase for biosensor applications with optimal activity around 85°C designed for acidic conditions around pH 5.0 The protein should maintain activity above 75°C. |
| 36 | I need to design a laccase for industrial biocatalysis applications designed to operate efficiently at 90°C featuring stabilized alpha-helical regions The protein should maintain activity above 65°C. |
| 37 | I want to create a biosensor protein for biotechnology research designed to operate efficiently at 70°C with disulfide bonds for enhanced structural stability It should maintain structural integrity under stress conditions. |
| 38 | I need to engineer an oxidase for diagnostic assay development capable of maintaining activity at 50°C that remains soluble at high concentrations with optimized hydrophobic core packing approximately 120 to 200 residues in length. |
| 39 | I need to engineer a glucosidase for textile industry applications designed to operate efficiently at 70°C with improved expression and solubility characteristics in the range of 80 to 160 residues. |
| 40 | I would like to engineer a peroxidase for detergent formulations that exhibits high solubility and minimal aggregation featuring mainly beta-sheet architecture with disulfide bonds for enhanced structural stability. |
| 41 | I want to engineer a peroxidase for detergent formulations with optimal activity around 55°C that exhibits high solubility and minimal aggregation The protein should have minimal aggregation tendency. |
| 42 | I need to engineer a phosphatase for agricultural applications that remains soluble at high concentrations approximately 100 to 250 residues in length The protein should be amenable to further engineering. |
| 43 | I'm looking to develop an isomerase for brewing and fermentation that can function at temperatures up to 65°C with enhanced solubility in aqueous solutions with broad pH tolerance from 4.0 to 7.0. |
| 44 | I need to develop an esterase for biotechnology research incorporating salt bridges for improved stability with predominantly alpha-helical secondary structure The design should avoid oxidation-sensitive residues where possible. |
| 45 | I need to design a laccase for agricultural applications with optimal activity around 60°C with disulfide bonds for enhanced structural stability that functions optimally at pH 6.0 comprising 80-230 amino acids. |
| 46 | I'm looking to develop a laccase for pharmaceutical manufacturing processes with high thermostability to withstand 90°C with predominantly alpha-helical secondary structure Long-term storage stability is important. |

*(continued on next page)*

*(continued from previous page)*

| ID | Requests |
|---|---|
| 47 | I'm looking to develop a protease for brewing and fermentation designed to operate efficiently at 65°C Long-term storage stability is important. The protein should be amenable to further engineering. |
| 48 | I require a ligase for textile industry applications that remains stable and active at 75°C with enhanced overall structural integrity designed for acidic conditions around pH 7.0 in the range of 80 to 230 residues with a mixed alpha-beta fold. |
| 49 | I need to engineer an amylase for waste treatment processes with high thermostability to withstand 80°C that remains soluble at high concentrations featuring stabilized alpha-helical regions with broad pH tolerance from 7.5 to 10.5. |
| 50 | I'm interested in creating a transferase for textile industry applications with excellent solubility properties for downstream processing stable across a pH range of 6.5 to 9.5 rich in alpha-helical content. |
| 51 | I want to engineer an amylase for agricultural applications with optimal activity around 55°C designed for acidic conditions around pH 6.0 Long-term storage stability is important. |
| 52 | I want to create a biosensor protein for cosmetic formulations capable of maintaining activity at 95°C with enhanced solubility in aqueous solutions incorporating salt bridges for improved stability. |
| 53 | I want to create a protease for textile industry applications capable of maintaining activity at 65°C with optimized hydrophobic core packing High expression yield in bacterial systems is preferred. |
| 54 | I would like to engineer a signaling protein for therapeutic drug development capable of maintaining activity at 65°C with broad pH tolerance from 4.0 to 7.0 featuring mainly beta-sheet architecture. |
| 55 | I require a dehydrogenase for waste treatment processes comprising 100-180 amino acids The protein should be amenable to further engineering. The protein should maintain activity above 75°C. |
| 56 | I need to develop a fluorescent protein for detergent formulations with optimal activity around 80°C with enhanced solubility in aqueous solutions featuring stabilized alpha-helical regions. |
| 57 | I need to develop a therapeutic protein for textile industry applications with optimal activity around 95°C incorporating salt bridges for improved stability The design should avoid oxidation-sensitive residues where possible. |
| 58 | I'm looking to develop a reductase for molecular biology applications with high thermostability to withstand 50°C that avoids aggregation and maintains solubility stable across a pH range of 4.5 to 7.5 with a length between 80 and 160 amino acids. |
| 59 | I want to engineer a laccase for biofuel production applications capable of maintaining activity at 60°C approximately 120 to 200 residues in length with stable beta-sheet regions. |
| 60 | I want to engineer a ligase for agricultural applications with high thermostability to withstand 70°C that remains soluble at high concentrations featuring stabilized alpha-helical regions with a length between 150 and 250 amino acids with stable beta-sheet regions. |
| 61 | I want to design a transporter protein for biosensor applications capable of maintaining activity at 55°C The design should avoid oxidation-sensitive residues where possible. that remains soluble at high concentrations. |
| 62 | I need to engineer a peroxidase for paper and pulp processing that exhibits high solubility and minimal aggregation with optimized hydrophobic core packing that functions optimally at pH 6.0 approximately 120 to 270 residues in length. |
| 63 | I need to engineer a cellulase for cosmetic formulations that remains stable and active at 55°C with improved expression and solubility characteristics with disulfide bonds for enhanced structural stability featuring mainly beta-sheet architecture High expression yield in bacterial systems is preferred. |

*(continued from previous page)*

| ID | Requests |
|----|----------|
| 64 | I need to engineer a cellulase for cosmetic formulations with optimal activity around 55°C with enhanced solubility in aqueous solutions The protein needs to be compatible with standard purification methods. |
| 65 | I'm interested in creating a biosensor protein for brewing and fermentation with high thermostability to withstand 95°C that avoids aggregation and maintains solubility with optimized hydrophobic core packing. |
| 66 | I would like to engineer a storage protein for chemical synthesis applications that can tolerate temperatures as high as 55°C with excellent solubility properties for downstream processing with enhanced overall structural integrity comprising 120-200 amino acids. |
| 67 | I need to design a biosensor protein for therapeutic drug development with high thermostability to withstand 70°C with enhanced overall structural integrity with a mixed alpha-beta fold. |
| 68 | I need to engineer a storage protein for pharmaceutical manufacturing processes with high thermostability to withstand 50°C in the range of 120 to 200 residues. |
| 69 | I would like to engineer a xylanase for biofuel production applications with enhanced solubility in aqueous solutions with a mixed alpha-beta fold The protein should maintain activity above 65°C. |
| 70 | I need to develop a biosensor protein for industrial biocatalysis applications that can function at temperatures up to 90°C with broad pH tolerance from 5.5 to 8.5. |
| 71 | I want to create a phosphatase for brewing and fermentation with optimal activity around 95°C with disulfide bonds for enhanced structural stability approximately 150 to 300 residues in length The protein needs to be compatible with standard purification methods. |
| 72 | I'm interested in creating a transferase for molecular biology applications that remains soluble at high concentrations with optimized hydrophobic core packing The protein needs to be compatible with standard purification methods. |
| 73 | I would like to engineer an industrial enzyme for molecular biology applications that can function at temperatures up to 70°C that exhibits high solubility and minimal aggregation with disulfide bonds for enhanced structural stability stable across a pH range of 7.5 to 10.5 in the range of 100 to 250 residues featuring mainly beta-sheet architecture. |
| 74 | I need to engineer a storage protein for chemical synthesis applications capable of maintaining activity at 65°C with enhanced solubility in aqueous solutions designed for acidic conditions around pH 4.0 comprising 150-230 amino acids. |
| 75 | I require a fluorescent protein for environmental remediation that can function at temperatures up to 90°C with excellent solubility properties for downstream processing It should maintain structural integrity under stress conditions. |
| 76 | I'm interested in creating a transporter protein for chemical synthesis applications with optimal activity around 95°C The design should avoid oxidation-sensitive residues where possible. that exhibits high solubility and minimal aggregation. |
| 77 | I need to develop a laccase for textile industry applications that remains stable and active at 60°C with improved expression and solubility characteristics incorporating salt bridges for improved stability that functions optimally at pH 5.5 in the range of 120 to 170 residues. |
| 78 | I need to design a binding protein for cosmetic formulations designed to operate efficiently at 70°C with improved expression and solubility characteristics with optimized hydrophobic core packing approximately 80 to 180 residues in length with stable beta-sheet regions Long-term storage stability is important. |
| 79 | I want to design a kinase for biofuel production applications with optimal activity around 75°C engineered for alkaline environments at pH 6.5 with improved expression and solubility characteristics. |

*(continued on next page)*

*(continued from previous page)*

| ID | Requests |
|---|---|
| 80 | I need to develop a reductase for brewing and fermentation that can function at temperatures up to 60°C The protein should maintain activity above 80°C. |
| 81 | I need to design a regulatory protein for textile industry applications that remains stable and active at 55°C featuring stabilized alpha-helical regions approximately 150 to 200 residues in length. |
| 82 | I need to design a ligase for industrial biocatalysis applications with high thermostability to withstand 95°C It should maintain structural integrity under stress conditions. The protein should have minimal aggregation tendency. |
| 83 | I want to engineer a lyase for chemical synthesis applications that can function at temperatures up to 55°C that exhibits high solubility and minimal aggregation with predominantly alpha-helical secondary structure. |
| 84 | I'm interested in creating an esterase for agricultural applications that remains stable and active at 95°C incorporating multiple stabilizing features that exhibits high solubility and minimal aggregation. |
| 85 | I need to design a protease for paper and pulp processing that remains soluble at high concentrations incorporating salt bridges for improved stability engineered for alkaline environments at pH 5.0. |
| 86 | I'm interested in creating a dehydrogenase for paper and pulp processing designed to operate efficiently at 90°C with improved expression and solubility characteristics incorporating multiple stabilizing features. |
| 87 | I want to design a kinase for environmental remediation with high thermostability to withstand 55°C that avoids aggregation and maintains solubility featuring mainly beta-sheet architecture. |
| 88 | I would like to engineer a lyase for therapeutic drug development that avoids aggregation and maintains solubility with optimized hydrophobic core packing The protein should have minimal aggregation tendency. |
| 89 | I need to develop a peroxidase for therapeutic drug development designed to operate efficiently at 65°C that avoids aggregation and maintains solubility comprising 120-220 amino acids High expression yield in bacterial systems is preferred. |
| 90 | I'm interested in creating a hydrolase for waste treatment processes incorporating salt bridges for improved stability engineered for alkaline environments at pH 7.5 The protein should have minimal aggregation tendency. |
| 91 | I want to create a xylanase for biosensor applications that remains stable and active at 90°C engineered for alkaline environments at pH 8.0 The design should avoid oxidation-sensitive residues where possible. |
| 92 | I would like to engineer an antimicrobial peptide for brewing and fermentation with disulfide bonds for enhanced structural stability It should maintain structural integrity under stress conditions. |
| 93 | I need to engineer a structural protein for therapeutic drug development with high thermostability to withstand 55°C with improved expression and solubility characteristics approximately 120 to 200 residues in length. |
| 94 | I'm looking to develop a signaling protein for agricultural applications that remains stable and active at 80°C that avoids aggregation and maintains solubility with disulfide bonds for enhanced structural stability. |
| 95 | I want to create a peptidase for therapeutic drug development that remains stable and active at 95°C that remains soluble at high concentrations with enhanced overall structural integrity featuring mainly beta-sheet architecture. |
| 96 | I want to design an antimicrobial peptide for biofuel production applications with high thermostability to withstand 90°C with disulfide bonds for enhanced structural stability with broad pH tolerance from 8.5 to 11.0 comprising 150-250 amino acids. |
| 97 | I'm looking to develop a hydrolase for diagnostic assay development approximately 150 to 200 residues in length with stable beta-sheet regions incorporating multiple stabilizing features. |
| 98 | I'm interested in creating a storage protein for paper and pulp processing that remains stable and active at 75°C with excellent solubility properties for downstream processing. |

*(continued on next page)*

| ID | Requests |
| --- | --- |
| 99 | I require a ligase for waste treatment processes with disulfide bonds for enhanced structural stability The protein should maintain activity above 70°C. that exhibits high solubility and minimal aggregation. |
| 100 | I want to create an enzyme inhibitor for detergent formulations with high thermostability to withstand 55°C with broad pH tolerance from 7.0 to 10.0 with a length between 80 and 130 amino acids. |

