# OpenReview forum: "PDAgent: An LLM-Driven Autonomous Agent Framework Towards *In Silico* Protein Design via Directed Mutation"
_ICML.cc/2026/Conference — ICML 2026 regular_

### Official Review · Reviewer_cPvA · 2026-02-16

**Soundness:** 2
**Presentation:** 3
**Significance:** 3
**Originality:** 2
**Overall Recommendation:** 4
**Confidence:** 3

**Summary:**

This paper presents PDAgent, an LLM-driven autonomous agent framework for computational protein design through template-based directed mutation. The system accepts natural language specifications of desired protein properties and employs a five-phase ReAct-style reasoning loop (THINK, PLAN, ACT, OBSERVE, REFLECT) to iteratively optimize protein sequences. The framework integrates template retrieval from UniProt, conservation-aware mutation strategies via BLAST/MSA, and a toolkit of seven domain-specific mutation operators targeting eight biophysical dimensions (thermostability, solubility, pH stability, length, structure preference, disulfide bonds, salt bridges). Experiments on 100 diverse protein design tasks show that PDAgent achieves 91.34% average constraint satisfaction rate with high structural quality (mean pLDDT 87.69), outperforming both direct LLM generation and specialized deep learning methods.

**Compliance With Llm Reviewing Policy:**

Affirmed.

**Final Justification:**

The authors provided thorough responses to all my concerns. The clarifications on benchmark design, backbone comparison, and conflict handling were convincing. I acknowledge the framework's extensibility and the soundness of the closed-loop design.

**Key Questions For Authors:**

1. How does PDAgent handle cases where no suitable templates are found in UniProt? What is the failure mode, and how often does this occur in the benchmark?

2. The thermostability CSR is substantially lower than other constraint types across all configurations. What are the primary reasons for this? Is the Tm predictor (PPTStab) a bottleneck?

3. How sensitive is the framework to the choice of LLM backbone for the reasoning components? The paper shows results with three backbones, but the performance differences (especially between LLaMA-3.1-8B and DeepSeek-V3.2) are not deeply analyzed.

4. How does the framework handle conflicting constraints (e.g., increasing thermostability while maintaining high solubility)? Are there systematic trade-offs observed?

**Limitations:**

No wet-lab experiments confirm whether designed proteins fold, express, or function as predicted. The gap between in silico predictions and experimental reality remains unaddressed.

**Strengths And Weaknesses:**

## Strengths

1. **Well-designed closed-loop optimization pipeline.** The five-phase ReAct loop with history memory is a principled approach to iterative protein refinement. The ablation study confirms each component's contribution, with the iteration loop being the most critical (removing it causes a 4.32-point average drop).

2. **Practical and accessible design paradigm.** Accepting natural language input significantly lowers the barrier to entry for non-expert users, bridging the gap between human intent and sequence-level optimization. This is a meaningful contribution over methods requiring predefined functional ontologies or 3D structural inputs.

3. **Conservation-aware mutation strategy.** Integrating evolutionary conservation analysis to protect functionally important residues during mutation is biologically well-motivated and adds a layer of safety to the optimization process. The conservation threshold analysis (Figure 3) provides useful guidance on this hyperparameter.

## Weaknesses

1. **Benchmark is self-generated and potentially biased.** The 100-task benchmark was generated using Claude Opus 4.5, which introduces circularity concerns — the tasks are designed by an LLM and evaluated on LLM-driven methods. There is no established community benchmark for comparison, and the task distribution may favor the proposed approach. The constraint types and parameter ranges may not reflect the true difficulty distribution of real-world protein design problems.

2. **No experimental validation.** All evaluations are purely computational (in silico). The designed proteins have not been synthesized, expressed, or experimentally characterized. Predicted properties (e.g., Tm from PPTStab, foldability from ESMFold) may not accurately reflect actual biophysical behavior, especially for heavily mutated sequences.

3. **Limited constraint taxonomy.** The framework covers only seven biophysical dimensions, omitting critical functional properties such as binding affinity, catalytic activity, substrate specificity, and expression yield. These are often the most important and challenging aspects of real protein design. The authors acknowledge this but it significantly limits practical applicability.

---

> ### Author Rebuttal · Authors · 2026-03-31
>
> We sincerely thank the reviewer for these valuable and constructive comments. We provide our response below point by point.
>
> **Re-Weakness 1**
>
> We thank the reviewer for raising this concern. However, Claude Opus 4.5 generated only task requests, not protein sequences or optimization strategies, and PDAgent uses different LLM backbones (LLaMA, Qwen, DeepSeek) for execution, so no information leaks from the generator. Beyond that, a constraint like "Tm≥80°C, GRAVY<0, length 150-300 aa" is equally challenging for both human and LLM. The tasks were also designed to reflect real-world design goals rather than artificially easy ones. The complete benchmark is provided in Appendix D, and we welcome community efforts to extend it further.
>
> **Re-Weakness 2 & Limitation**
>
> We acknowledge this limitation, which is stated explicitly in Discussion section. We note that our computational tools are widely tested and trusted in the protein engineering community, providing reliable assessment. Our template-based approach using curated UniProt sequences also provides a strong starting point. Given the substantial time and cost of wet-lab experiments, computational screening serves as a critical step before experimental testing. We will strengthen this discussion in the revised paper.
>
> **Re-Weakness 3**
>
> We agree that the current framework covers only a subset of dimensions for practical protein design. The seven dimensions were chosen because they: (1) can be evaluated directly from sequence alone, keeping per-iteration runtime at ~45 seconds; (2) represent basic requirements for any functional protein, since stability, solubility, and proper folding must be satisfied before binding or catalytic activity become relevant. Our main contribution is the closed-loop agent framework itself, which is extensible: adding new constraint types only requires defining the evaluation function and mutation tool, with no changes to the architecture.
>
> **Re-Question 1**
>
> We thank the reviewer for this question. PDAgent employs a progressive fallback retrieval strategy, with implementation available in our anonymous repository (`agent/coordinator.py`, lines 118–131). Specifically, the system generates keyword groups from the most specific to the least. If the full query (e.g., "A AND B AND C") returns insufficient results, it progressively simplifies (e.g., "A AND B", then "B") until enough templates are retrieved. Across all 100 benchmark tasks, we successfully retrieved suitable templates for all tasks. We will add it as pseudocode in the revised paper.
>
> **Re-Question 2**
>
> We thank the reviewer for this question. We analyzed the target Tm distribution across all thermostability tasks: 95% require Tm≥55°C, 52.5% require Tm≥70°C, and 21.25% require Tm≥90°C. Reaching such high targets by point mutations alone is inherently difficult. We believe PPTStab itself is not the main bottleneck, as it achieves a Pearson r of 0.89 and R² of 0.80 against experimental Tm values [1], indicating reliable prediction. The lower Thermo. CSR therefore reflects the physical difficulty of the targets themselves. Despite this, PDAgent still achieves superior performance. We will add this analysis in the revised paper.
>
> **Re-Question 3**
>
> We thank the reviewer for this valuable question. As a supplement to Table 1, we conducted a detailed comparison across the three backbones. We also split our benchmark into 76 non-conflicting and 24 conflicting tasks (detailed in Question 4) to make comparison:
>
> |Backbone|Fold.|Avg. CSR (%)|Avg. #Mut|Conflict CSR (%)|Non-conflict CSR (%)|Avg. #Iter|
> |-|-|-|-|-|-|-|
> |LLaMA-3.1-8B|87.69|90.60|6.05|84.69|92.64|3.14|
> |Qwen-3-30B|80.54|85.09|2.00|75.01|88.84|1.37|
> |DeepSeek-V3.2|87.69|91.34|13.02|89.12|92.93|10.41|
>
> The results reveal: (1) Qwen-3-30B's lower performance is mainly due to early stopping (1.37 iterations and 2.0 mutations per task), suggesting its REFLECT phase stops optimization too aggressively; (2) LLaMA-3.1-8B reaches similar performance to DeepSeek-V3.2 with fewer iterations, showing smaller models can work well within our framework; (3) The performance gaps are larger on conflicting tasks, confirming stronger LLM reasoning is most valuable when trade-offs are involved. We will add this analysis in the revised paper.
>
> **Re-Question 4**
>
> As shown in the Q3 table above, we identified 24 tasks with conflicting constraints (e.g., high thermostability+high solubility). PDAgent handles these well (89.12% CSR), though with an expected drop compared to non-conflicting tasks. The reasoning outputs show: the REFLECT phase detects when improving one constraint consistently worsens another and adjusts the strategy accordingly, while the THINK phase diagnoses the conflict and recommends a balanced approach.
>
> **Reference**
>
> [1] Tijare P, et al. Prediction and design of thermostable proteins with a desired melting temperature. Scientific Reports, 2025.

---

> > ### Author Rebuttal · Reviewer_cPvA · 2026-04-01
> >
> > I thank the authors for their detailed response. After carefully reading the rebuttal, I maintain my score.

---

> > > ### Author Response · Authors · 2026-04-02
> > >
> > > We thank the reviewer for carefully reading our rebuttal. We will incorporate all the discussed improvements in the revision.

---

### Official Review · Reviewer_tgbq · 2026-03-08

**Soundness:** 3
**Presentation:** 3
**Significance:** 3
**Originality:** 3
**Overall Recommendation:** 4
**Confidence:** 4

**Summary:**

This paper describes an AI agent method, PDAgent, to perform template mutation-based protein design according to the textual specification of protein properties provided by users. The work develops a five-phase automated reasoning and action framework to design proteins. The conservation analysis on multiple sequence alignments is used to improve the structural and functional integrity of designed proteins. A dataset of 100 text-based designed tasks is curated. PDAgent performs better on the dataset than standard LLMs and deep learning-based methods of generating proteins from texts.

**Compliance With Llm Reviewing Policy:**

Affirmed.

**Key Questions For Authors:**

(1) Could you explain why BLAST, MAFFT, DDGun  and PPTStab are selected for conservation analysis, mutation proposal, and stability prediction?

**Limitations:**

Yes.

**Strengths And Weaknesses:**

Strengths:

(1) PDAgent is a fully automated, closed-loop pipeline consisting of design, computational validation and refinement steps. It is different from the existing methods that typically have only one generation step.

(2) PDAgent performs better than the baseline methods in the proposed experimental setup.


Weaknesses:

(1)  The foldability of proteins is assessed using pLDDT score of  ESMFold. However, ESMFold is not the state-of-the-art protein structure prediction method. Some evaluation results based on the state-of-the-art method such as AlphaFold3 should be included.

(2) Using pLDDT score of predicted structures of designed proteins to assess their quality favors proteins whose sequences are very similar to known natural proteins in the protein database such as UniProt but penalizes novel proteins whose sequences are not similar to the existing proteins. However, designing novel proteins is also an important objective of protein design. Some metrics of evaluating the novelty and diversity of designed proteins should be added into the evaluation.

(3) The proposed method is compared with standard LLMs and some deep learning-based protein design methods. However, it does not compare it with a simple, yet relevant baseline method - the template-based approach that uses templates retrieved from UniProt based on key words as designed proteins. This comparison on the test dataset is necessary to assess how much the proposed approach improves over start templates.

(4) Figure 4 does not show how one of targeted constraints (pH value) changes over iterations.

---

> ### Author Rebuttal · Authors · 2026-03-31
>
> We sincerely thank the reviewer for these valuable and constructive comments. We provide our response below point by point.
>
> **Re-Weakness 1**
>
> We agree that AlphaFold3 is more accurate and re-evaluated a random 30-sequence subset of designed proteins using AlphaFold3 (via the AlphaFold Server):
>
> |Model|ESMFold pLDDT|AlphaFold3 pLDDT|Pearson Correlation|
> |-|-|-|-|
> |PDAgent|87.85|91.00|0.92|
> |Pinal|86.91|87.23|0.88|
> |Direct LLM (Gemini)|52.32|56.60|0.91|
>
> As shown in the table, the strong correlation confirms that ESMFold provides reliable comparative evaluation. We chose ESMFold for in-loop evaluation because AlphaFold3 requires ~180s per sequence (vs. ESMFold's ~1.92s), making it too slow for real-time feedback during iterative optimization. We will add this result in the revised paper and discuss the trade-off between prediction accuracy and computational efficiency.
>
> **Re-Weakness 2**
>
> We thank the reviewer for this valuable suggestion. We computed novelty and diversity metrics across all refined sequences:
>
> |Metric|PDAgent|Pinal|Direct LLM (Gemini)|
> |-|-|-|-|
> |Mean identity to UniProt hit (%)|95.96|61.88|10.99|
> |Intra-set pairwise diversity (%, 1−mean pairwise identity)|75.28|78.18|70.82|
>
> PDAgent's higher sequence identity to UniProt is intentional: our template-based approach deliberately conserves known functional scaffolds to ensure structural reliability, which is the core premise of our method. In contrast, sequences generated directly by LLMs are highly novel but show poor structural quality, precisely because they lack the evolutionary grounding that natural proteins have. There is a trade-off between novelty and reliability, and PDAgent prioritizes reliability which we believe is more appropriate for practical protein engineering. We will add this result and discussion in the revised paper.
>
> **Re-Weakness 3**
>
> This is an important missing comparison. We evaluated the best-retrieved UniProt template before any mutation as a baseline:
>
> |Method|CSR (%)|Fold.|
> |-|-|-|
> |Best template|82.24|88.68|
> |PDAgent (LLaMA-3.1-8B)|90.60|87.69|
> |Δ|+8.36|−0.99|
>
> PDAgent consistently improves CSR over unmodified templates across all backbones (+8.36 for LLaMA-3.1-8B, +19.21 for Qwen-3-30B, +6.13 for DeepSeek-V3.2), confirming that the optimization loop contributes substantial value beyond simply retrieving a good template. The slight pLDDT decrease reflects necessary mutations made to satisfy design constraints, consistent with our discussion about the trade-off in Section 4.3. We will add these baselines to Table 1 in the revised paper.
>
> **Re-Weakness 4**
>
> We thank the reviewer for pointing this out. We clarify that the pH stability constraint is already shown in Figure 4 as the isoelectric point (pI) trajectory. As defined in Section 3.1 and Appendix Table 4, pH stability is assessed through pI matching. To avoid this confusion, we will add a clearer explanation in the revised paper.
>
> **Re-Question 1**
>
> We thank the reviewer for this question. Here is our rationale for each tool:
>
> - BLAST[1]: It is the most widely used and computationally efficient tool for homolog searching, and we use it against SwissProt for speed and reproducibility.
>
> - MAFFT[2]: It is one of the most widely used MSA tools, achieving a good balance between quality and speed, while alternatives like MUSCLE and Clustal Omega either sacrifice speed for accuracy or scalability for simplicity.
>
> - DDGun[3]: It is a well-established physics-inspired method that predicts the effect of a mutation on protein stability without needing training data from the specific protein, making it applicable across different protein families.
>
> - PPTStab[4]: It is a recently published method that predicts melting temperature directly from sequence without requiring a 3D structure as input, and has shown superior accuracy against experimental results, indicating reliable prediction.
>
> We acknowledge that alternative tools exist for each component, and PDAgent's modular design allows straightforward substitution. We have discussed this extensibility in the Conclusion section.
>
> **Reference**
>
> [1] Camacho C, et al. BLAST+: architecture and applications. BMC Bioinformatics, 2009.
>
> [2] Katoh K, et al. MAFFT multiple sequence alignment software version 7: improvements in performance and usability. Molecular Biology and Evolution, 2013.
>
> [3] Montanucci L, et al. DDGun: an untrained method for the prediction of protein stability changes upon single and multiple point variations. BMC Bioinformatics, 2019.
>
> [4] Tijare P, et al. Prediction and design of thermostable proteins with a desired melting temperature. Scientific Reports, 2025.

---

> > ### Author Rebuttal · Reviewer_tgbq · 2026-04-01
> >
> > The authors provided new data or rationale to address all my concerns.

---

> > > ### Author Response · Authors · 2026-04-03
> > >
> > > We appreciate the reviewer's time in reading our rebuttal. We will incorporate all the discussed improvements in the revision.

---

### Official Review · Reviewer_WKam · 2026-03-11

**Soundness:** 3
**Presentation:** 3
**Significance:** 3
**Originality:** 3
**Overall Recommendation:** 4
**Confidence:** 3

**Summary:**

PDAgent frames natural language-guided protein design as a constraint satisfaction problem and addresses it by retrieving UniProt templates and iteratively applying rule-based mutations through a five-phase ReAct loop. The core loop calls a toolkit of seven specialized mutation operators, with conserved positions protected via MSA-based analysis and individual mutations filtered by a stability predictor. On a benchmark of 100 design tasks, PDAgent substantially outperforms both direct LLM generation and specialized deep learning methods on average constraint satisfaction while maintaining high predicted structural quality.

**Compliance With Llm Reviewing Policy:**

Affirmed.

**Final Justification:**

The authors have resolved my concern in the rebuttal.

**Key Questions For Authors:**

1. The abstract mentions eight biophysical dimensions while the method section defines seven. What is the correct count?

2. The case study template already satisfies the thermostability constraint at iteration 0, yet thermostability is where PDAgent performs worst overall. Can the authors show a case where this constraint was initially unmet and the system successfully optimized it?

3. What is the distribution of sequence identity between the final designs and their initial templates? This would clarify wether the system performs substantive redesign or predominantly makes a small number of localized edits.

**Strengths And Weaknesses:**

## Strengths

- The per-dimension breakdown reveals structurally different failure modes: specialized deep learning methods achieve competitive structural quality but largely fail at multi-constraint satisfaction, while direct LLM generation shows the opposite pattern. PDAgent is the only approach that performs well on both dimensions simultaneously.

- The ablation study identifies iterative refinement as the dominant contributor, with the iteration loop removal causing the largest single performance drop. The candidate-count analysis independently corroborates this, showing constraint satisfaction improving substantially as template candidates increase.

## Weaknesses

- The conservation guard is presented as a core contribution for structural integrity, but removing it slightly increases predicted structural quality while the average metric barely changes, which is the opposite of what the mechanism predicts. The ablation section does not address this contradiction specifically.

- The comparisons do not isolate the contribution of LLM reasoning from iterative tool use. All baselines are single-pass generators, so the gap shows iteration helps, but not that LLM diagnosis is necesary over a simpler iterative policy that selects mutation operators based on which constraints are currently violated, without any LLM reasoning.

- The efficiency analysis uses a single-candidate setup, while the headline results run three independent optimization chains and select the best. Since the LLM calls are stochastic, running the same template three times and taking the best would likely show a similar improvment, so it is unclear how much of the gain comes from template diversity versus oracle selection over multiple runs.

---

> ### Author Rebuttal · Authors · 2026-03-31
>
> We sincerely thank the reviewer for these valuable and constructive comments. We provide our response below point by point.
>
> **Re-Weakness 1**
>
> We thank the reviewer for this insightful comment. The slight pLDDT increase upon removing the conservation guard reflects a key distinction between structural confidence and functional conservation. The guard protects positions critical to function (e.g., active sites, binding sites), as identified by evolutionary analysis. Many such positions exist in naturally flexible regions where ESMFold assigns lower confidence scores. Removing the conservation guard allows mutations at these positions, which can produce sequences that appear more "foldable" to ESMFold while potentially damaging biological function. To quantify this, without the guard, 68.5% of mutations fall at conserved positions, compared to 0% with it enabled, confirming the guard effectively keeps mutations away from functionally sensitive sites. We will add this analysis in the revised paper.
>
> **Re-Weakness 2**
>
> This is an important concern. We implemented a rule-based iterative controller using the same pipeline, tools, and templates, but replacing LLM reasoning with a simple policy that selects mutation operators based on which constraints are currently violated. We identified 24 tasks with conflicting constraints (e.g., high thermostability+high solubility) and reported CSR for both the rule-based controller and PDAgent:
>
> |Setting|Avg. CSR (%)|Conflict CSR (%)|Non-conflict CSR (%)|
> |-|-|-|-|
> |PDAgent|91.34|84.69|92.64|
> |PDAgent w/o LLM|88.55|71.77|91.72|
> |PDAgent w/o iteration|81.85|69.73|84.70|
>
> As shown, both iteration and LLM reasoning contribute to performance. The overall gap is 2.79 points, but the LLM's contribution concentrates on conflicting tasks (+12.92 points), where fixed rules cannot dynamically balance competing objectives. This confirms that iteration is the key contributor, while LLM reasoning is essential for handling trade-offs in complex scenarios. We will add this result in the revised paper.
>
> **Re-Weakness 3**
>
> We thank the reviewer for raising this point. To quantify how much of the gain comes from template diversity versus multiple runs, we ran a controlled experiment: we optimized the same single template 3 times independently and selected the best, mimicking multiple runs without using diverse templates.
>
> |Setting|CSR (%)|Fold.|
> |-|-|-|
> |PDAgent, n=3 diverse templates (headline)|90.60|87.69|
> |Single template×3 runs, oracle selection|78.33|83.20|
> |Single template×1 run|74.51|83.06|
>
> The results show that oracle selection among multiple runs adds only 3.82 points (78.33 vs. 74.51), while template diversity adds 12.27 points (90.60 vs. 78.33). This confirms that the performance gain mainly comes from diverse starting points rather than multiple stochastic runs, validating our design choice. We will add this analysis in the revised paper.
>
> **Re-Question 1**
>
> We apologize for this confusion. The correct count is 7 constraint types, as defined in Section 3.1. The abstract mistakenly includes foldability as the eighth dimension. However, foldability serves as a metric to assess the structural quality of designed proteins, rather than a user-specified design constraint. We will correct this in the revised paper.
>
> **Re-Question 2**
>
> We thank the reviewer for this question. We provide an additional case where thermostability starts unmet.
>
> > Task (Task 79 in Table 6): *"Design a kinase for biofuel production applications with optimal activity around 75°C, engineered for alkaline environments at pH 6.5, with improved expression and solubility characteristics."*
>
> Retrieved template: UniProt B9K712. Constraints: Tm≥75°C, pH≈6.5, soluble.
>
> |Iter.|Tm (°C)|GRAVY|pI|CSR|Mutations|
> |-|-|-|-|-|-|
> |0|73.64|−0.60|9.50|33%|—|
> |1|74.19|−0.61|9.08|33%|K9E, K10E, C64S, N74D|
> |2|74.20|−0.63|8.56|33%|R16E, G58P, F143Y|
> |3|74.96|−0.63|5.85|33%|K19E, Q59E|
> |4|75.44|−0.66|6.50|100%|S30P, L56K, S64C|
>
> The template initially fails the thermostability constraint. Over 4 iterations, PDAgent gradually refines the protein using `enhance_thermostability` and `adjust_pI` tools and finally satisfies all constraints. We believe the relatively low overall Thermo. CSR is mainly due to the demanding target distribution in our benchmark (95% require Tm≥55°C, 52.5% require Tm≥70°C). We will add this case study in the revised paper.
>
> **Re-Question 3**
>
> We analyzed sequence identity between all refined sequences and their initial templates. The results show a mean identity of 92.81% (std=6.17%), ranging from 74.19% to 98.20%, with an average of 8.07 (std=5.00) mutations per sequence. This confirms that PDAgent makes targeted, localized edits at non-conserved positions rather than substantive redesign. Notably, these results show that a small number of well-chosen point mutations can substantially improve biophysical properties while preserving the functional scaffold. We will include this result in the revised paper.

---

> > ### Author Rebuttal · Reviewer_WKam · 2026-04-02
> >
> > I thank the authors for the detailed reply.

---

> > > ### Author Response · Authors · 2026-04-03
> > >
> > > We sincerely thank the reviewer for the positive update and for the valuable feedback throughout the review process. We will incorporate all the discussed improvements in the revision.

---

### Official Review · Reviewer_jTrU · 2026-03-13

**Soundness:** 3
**Presentation:** 3
**Significance:** 2
**Originality:** 2
**Overall Recommendation:** 4
**Confidence:** 3

**Summary:**

This paper introduces PDAgent, an LLM-driven autonomous agent framework for protein design. PDAgent first parses a user’s natural-language query of desired protein properties to extract query terms; with these, PDAgent then retrieves related existing protein sequences from UniProt as realistic starting templates. For each template, PDAgent determines critical residues to preserve via conservation analysis with BLAST and multiple sequence alignment. Then, PDAgent iteratively makes point mutations via a ReAct-style 5-stage closed optimization loop, integrating LLM reasoning with a toolkit of 7 mutation operators and computational biology evaluation tools (ESMFold). The paper also provides a benchmark of 100 diverse design tasks generated by Claude Opus 4.5 spanning different enzyme classes and constraint combinations; PDAgent with DeepSeek‑V3.2 achieved 91.34% average on the paper’s constraint satisfaction rate (CSR) metric, substantially outperforming both direct LLM generation and specialized deep learning methods like ProtDAT.

**Compliance With Llm Reviewing Policy:**

Affirmed.

**Final Justification:**

After the rebuttal, the soundness of the methodology is boosted by the rule-based controller comparison, transparency on the baseline of unmodified best templates, cross-validation with AlphaFold3, and reported failure cases.

I think there are still core tensions that are unresolved in lack of external validation, so the rebuttal moves my assessment to borderline accept/reject; I would not be opposed to any other reviewers’ recommendations to accept or reject the paper.

**Key Questions For Authors:**

1. Does LLM reasoning outperform less trivial baselines (e.g. a rule-based controller) on the same tools? On the exact same pipeline, how does replacing the LLM reasoning with a simple deterministic strategy compare? If a simple greedy optimizer on the same tools matches PDAgent's performance, does the LLM add any value?
2. How does PDAgent perform on an established protein engineering benchmark? Some examples are ProteinGym stability prediction tasks or the Rocklin et al. mini-protein dataset. This would greatly address the self-referentiality of the presented results, as well as possibly allow for experimental comparisons if such data exists.
3. Do any of the proteins designed by PDAgent fold and function experimentally?
4. What happens when constraints (e.g. thermostability vs. flexibility) conflict? High performance on a set of tasks with deliberately conflicting constraints would greatly strengthen the core claim that LLM-guided decision-making adds value to scientific optimization.
5. Are the satisfaction thresholds so loose that random mutations could achieve high CSR? The presented random baseline value (55.46%) is rather high.

**Limitations:**

The limitations discussion currently omits the most important methodological limitations (evaluation circularity, unknown LLM contribution, benchmark narrowness).

The societal impact statement is minimal; though dual-use risk is likely negligible as-is because the system does not handle any functionally dangerous properties, additional comments should be made about potential impact as the framework extends toward more powerful capabilities. An explicit comment that computational designs require experimental validation before deployment is likely useful in this section as well.

**Strengths And Weaknesses:**

Soundness: 2
- Strengths
  - The system is correctly implemented and does what it claims at the pipeline level.
  - The math and logic behind the paper holds. The algorithm is correctly specified, and the ablations are correctly executed.
- Weaknesses
  - The benchmark relies on summary metrics and proxy objectives that are biased towards the framework’s strengths and can be misleading themselves.
    - The evaluation may be circular. The heuristics behind the selected set of mutations also directly optimize the same properties measured by the evaluation metrics, so high CSR is largely a consequence of applying transforms and checking their algebraic effects.
    - Constraint taxonomy is proxy-heavy (e.g., “pH stability” ≈ pI matching within ±2, solubility ≈ GRAVY≤0). These may be easy to game.
    - Starting from known structures as templates, while accurately capturing realistic structures, may distort how effective PDAgent is by beginning from a point where CSR metrics are already high.
    - Average CSR as reported is presented as an overall constraint satisfaction score but may be effectively a macro-average over constraint types (e.g. Thermo) instead. This can dramatically distort performance under the paper’s highly imbalanced constraint frequencies.
  - Baseline comparisons are weak.
    - Direct LLM prompts and specialized generation methods are being evaluated on a task they weren't designed for.
    - A baseline using a simple rule-based controller instead of LLM reasoning for the same tools, templates, and conservation analysis, which would isolate the LLM's contribution, would greatly strengthen the baseline comparisons.
  - Experimental validation is missing, though this is acknowledged.
  - The ablation study reveals that the LLM-specific components (history memory, THINK/PLAN separation) contribute small and possibly insignificant improvements (0.72 and 1.97 points respectively, with no variance reported).

Presentation: 3
- Strengths
  - The paper is readable and flows logically.
  - The pipeline is easy to grasp. The central overview figure (Figure 1) is helpful.
  - The paper provides enough detail to reproduce the system in principle.
- Weaknesses
  - The presented case study reads like a cherry-picked success case that may not be representative of all cases. More analysis on areas where the framework struggles or fails would be helpful.
  - Discussion is thin. More discourse on lessons learned and what the results mean for the broader landscape would strengthen this section.
  - Positioning against related work is relatively weak. Discussion of how exactly PDAgent addresses gaps in prior work would be helpful, especially in differentiating between the most closely related work (ProtAgents, Biomni, DrugPilot). Similarly, comparisons to existing “classical” protein engineering workflows that do not involve agents (e.g. Rosetta) would also be helpful.

Significance: 2
- Strengths
  - The problem domain is important.
  - The argument of enabling non-experts to prototype protein designs has merit.
  - The system could in principle influence future work as a scaffolding that others extend with better tools, harder constraints, and experimental validation.
- Weaknesses
  - The paper’s contribution is diminished in optimizing coarse proxy metrics via conservative template mutations.
  - The contribution (applying ReAct to a new domain) is thin when it comes to contributing a meaningful methodological advance.
  - Practical utility is completely unevaluated, e.g. in downstream use, user studies, comparison against existing non-agent expert workflows.

Originality: 2
- Strengths
  - The synthesis of PDAgent’s components (ReAct-style reasoning, LLM agents for protein design, template-based directed mutation) into a closed-loop system with formal constraint satisfaction is novel.
  - The presented benchmark is novel, though not in methodology; it may also be constrained by its specific applicability to PDAgent.
- Weaknesses
  - All components of PDAgent are well-established in the literature.
  - The paper does not demonstrate that the specific combination of components yields capabilities that simpler alternatives cannot match.
  - The paper only shallowly distinguishes PDAgent and closely related work (ProtAgents, Biomni, DrugPilot); more depth in discussion is needed.

---

> ### Author Rebuttal · Authors · 2026-03-31
>
> We sincerely thank the reviewer for these valuable and constructive comments. We provide our response below point by point.
>
> **Re-Weakness-Soundness**
>
> **W1-1:** We cross-validated 30 designs using AlphaFold3: ESMFold pLDDT=87.85, AlphaFold3 pLDDT=91.00, Pearson r=0.92, confirming our metrics are not circular artifacts.
>
> **W1-2:** These are standard biophysical heuristics widely used in protein engineering. While individual proxies appear simple, satisfying multiple constraints while maintaining high pLDDT is non-trivial, as evidenced by frontier baselines in Table 1.
>
> **W1-3:** We evaluated unmodified best templates as a baseline: template CSR=82.24%, Fold.=88.68. PDAgent improves CSR by 8.36 points, confirming optimization contributes substantial value beyond template retrieval.
>
> **W1-4:** We now report micro-average CSR: PDAgent 82.08%, Gemini 64.68%, Pinal 43.70%. Relative rankings remain consistent. We will report them in the revision.
>
> **W2-1:** This highlights that PDAgent fills a gap where no existing method targets natural language (NL)-driven multi-constraint protein optimization. The specialized methods are already the closest text-conditioned approaches available.
>
> **W2-2:** We implemented a rule-based controller using the same pipeline but replacing LLM with a simple policy. PDAgent outperforms it by 2.79 points overall, with the gap concentrating on 24 conflicting-constraint tasks (84.69% vs. 71.77%), confirming LLM reasoning is essential for trade-off handling. We will add this result in the revision.
>
> **W3:** We acknowledge this limitation, as discussed. Our method builds on validated computational tools and curated UniProt templates. Given the costly wet-lab validation, computational screening is a critical precursor.
>
> **W4:** We now report variance over 3 runs: History Memory (std=0.51) and THINK/PLAN Sep. (std=1.36). History Memory contributes more on complex tasks, while the variance of THINK/PLAN Sep. suggests its impact is task-dependent.
>
> **Re-Weakness-Presentation**
>
> **W1:** We provide a failure case: Task 49 (target Tm≥80°C, template Tm=69.3°C) reached only 72.2°C after 6 iterations with CSR improving from 50% to 75%, due to demanding target constraints.
>
> **W2:** We will expand to include lessons learned and broader implications for agentic scientific design.
>
> **W3:** ProtAgents lacks closed-loop validation; Biomni lacks structured optimization cycles; DrugPilot targets drug molecules, not protein; Rosetta requires expert setup. PDAgent uniquely combines NL input with iterative optimization. We will add these comparisons in the revision.
>
> **Re-Weakness-Significance**
>
> **W1&W2:** While the current proxy metrics seem coarse, they are modular and can be easily replaced with complex alternatives. As for methodology, adapting ReAct to protein design is not trivial, as it required formalizing NL into structured constraints, designing mutation strategies, and implementing closed-loop optimization. To our knowledge, PDAgent is the first automated closed-loop system for NL-driven protein design.
>
> **W3:** We note that PDAgent completes each design in ~2.3 mins via NL input, greatly lowering the barrier for non-expert users compared to tools like Rosetta. We will discuss this as future work in the revision.
>
> **Re-Weakness-Originality**
>
> **W1:** While individual components are established, their integration into a closed-loop NL-driven protein design system is novel and required substantial domain adaptation.
>
> **W2:** The rule-based controller experiment (*Soundness W2-2*) shows that PDAgent outperforms an identical pipeline with deterministic reasoning, confirming the LLM-driven combination adds capabilities beyond simpler alternatives.
>
> **W3:** Please refer to *Presentation W3*.
>
> **Re-Question**
>
> **Q1:** Please refer to *Soundness W2-2*. It shows that LLM reasoning adds clear value in conflict handling and adaptive decision-making over simple greedy optimization.
>
> **Q2:** Existing benchmarks like ProteinGym (single-mutation fitness) and Rocklin (de novo mini-proteins) target fundamentally different tasks from PDAgent. Directly applying PDAgent to them would not lead to fair comparisons. We will discuss the need for standardized NL-driven design benchmarks in the revision.
>
> **Q3:** Not yet. Please refer to *Soundness W3*.
>
> **Q4:** We identified 24 tasks with conflicting constraints, and PDAgent achieves 84.69% CSR on these vs. 92.64% on non-conflicting tasks. The reasoning traces show the agent rationally navigates these trade-offs through prioritization and compromise strategies.
>
> **Q5:** Although random generation reaches 55.46% CSR, its pLDDT of only 30.24 indicates poor structural quality. Other methods like BioMedGPT-10B achieve far lower CSR (28.69%), confirming the thresholds are not loose. Achieving both high CSR and structural quality is the real challenge, which only PDAgent accomplishes.
>
> **Re-Limitation:** We will revise the Limitations section to explicitly include these points.

---

> > ### Author Rebuttal · Reviewer_jTrU · 2026-04-04
> >
> > I thank the authors for their detailed and thoughtful reply. The soundness of the methodology is boosted by the rule-based controller comparison, transparency on the baseline of unmodified best templates, cross-validation with AlphaFold3, and reported failure cases.
> >
> > I appreciate the answers to key questions, particularly the clarifications around related work and benchmarks.
> >
> > I think there are still core tensions that are unresolved in lack of external validation, so the rebuttal moves my assessment to borderline accept/reject; I would not be opposed to any other reviewers’ recommendations to accept or reject the paper.

---

> > > ### Author Response · Authors · 2026-04-05
> > >
> > > We appreciate the reviewer's time and constructive feedback throughout the review process. We will incorporate all discussed improvements, including external validation discussion, in the revision.

---

### Decision · Program_Chairs · 2026-04-30

**Decision:**

Accept (regular)

**Comment:**

Reviewers consistently agree that the submission addresses a significant problem in the field. The proposed method is technically sound, the experimental evaluations are comprehensive.

Reviewers also raised some minor questions, some of which have been well addressed during the rebuttal period. We suggest that the authors revise the paper accordingly to produce an improved final version.